# Learning with Labeling Induced Abstentions

**Kareem Amin**
Google Research
New York, NY
kamin@google.com

**Giulia DeSalvo**
Google Research
New York, NY
giuliad@google.com

**Afshin Rostamizadeh**
Google Research
New York, NY
rostami@google.com

## Abstract

Consider a setting where we wish to automate an expensive task with a machine learning algorithm using a limited labeling resource. In such settings, examples routed for labeling are often out of scope for the machine learning algorithm. For example, in a spam detection setting, human reviewers not only provide labeled data but are such high-quality detectors of spam that examples routed to them no longer require machine evaluation. As a consequence, the distribution of examples routed to the machine is intimately tied to the process generating labels. We introduce a formalization of this setting, and give an algorithm that simultaneously learns a model and decides when to request a label by leveraging ideas from both the abstention and active learning literatures. We prove an upper bound on the algorithm's label complexity and a matching lower bound for any algorithm in this setting. We conduct a thorough set of experiments including an ablation study to test different components of our algorithm. We demonstrate the effectiveness of an efficient version of our algorithm over margin sampling on a variety of datasets.

## 1 Introduction

In this paper, we consider a system that relies on automated predictions made by a machine learning model. We assume this system has a limited budget for requesting ground-truth labels (e.g. from a domain expert). In practice, such request can be used, among other purposes, to gather additional training data for the machine learning model. If the system asks for a label for an example, then it no longer needs the model's prediction for that particular example. Thus, the pattern of label queries effectively defines the distribution that the model will learn with and predict on.

Take for example a large-scale video-hosting website. The website wants to automatically detect videos that violate its community guidelines. In order to acquire labels for this task, some of these videos are evaluated by a finite pool of human reviewers. There are two consequences of this evaluation. Firstly, the reviewer provides a training label for the model. Secondly, the human intervention makes it so that the machine learning algorithm is no longer tasked with making predictions on these examples. As a result, the goal is then optimize the the model's performance in the domain where it will be executed.

We take the perspective of a system designer who wants to understand (and optimize for) the performance of the automated system on the examples that it will be asked to evaluate. Let $r : \mathcal{X} \to \{0, 1\}$ be a rule governing whether the system requests a label for $x \in \mathcal{X}$. Let $h : \mathcal{X} \to \mathbb{R}$ be a hypothesis describing the automated system. We seek to minimize $\mathbb{E}[L(h(x), y) \mid r(x) = 0]$, where $L$ is a loss function. We can think of this setting as combining the objective studied in *abstention learning* (minimizing $E[L(h(x), y) \mid r(x) = 0]$) with the feedback model studied in *active learning* (labels are only available when $r(x) = 1$). We introduce a new framework, which we call *dual purpose learning* framework, that combines these elements.

35th Conference on Neural Information Processing Systems (NeurIPS 2021).

We first analyze the *proper* dual purpose labeling framework in an online setting, which proceeds as follows. At the start of each round $t$, the learner selects both a requester function $r_t$ from some class $\mathcal{R}_\rho$, where $\rho$ is an upper bound on the request-rate of functions in this class, and a hypothesis $h_t$ from some class $\mathcal{H}$ using all the feedback available from the past. The learner's expected loss on this round is given by $E[L(h_t(x), y) \mid r_t(x) = 0]$. The function $r_t$ also determines the feedback available to the algorithm. If $r_t(x) = 1$, the learner observes $(x, y)$, which was drawn i.i.d. from an unknown distribution, and otherwise, only $x$ is revealed and $y$ is censored. The goal of the learner is to compete with the optimal choice of hypothesis and requester by minimizing the excess loss $\mathcal{L}_{\mathcal{H}, \mathcal{R}_\rho}(h_t, r_t) = E[L(h_t(x), y) \mid r_t(x) = 0] - \inf_{(h^*, r^*) \in \mathcal{H} \times \mathcal{R}_\rho} \mathbb{E}[L(h^*(x), y) \mid r^*(x) = 0]$.

Our first main result is the surprising fact that, under mild assumptions, bounds on the excess loss that match the $O(1/\sqrt{t})$ generalization rates of full-feedback passive learning are not possible in the proper dual purpose labeling framework. This lower bound also suggests a relaxation of the proper dual purpose labeling framework, which we call *improper dual purpose labeling*. In the improper setting, the learner is still interested in learning $h_t, r_t$ that minimize $\mathcal{L}_{\mathcal{H}, \mathcal{R}_\rho}$. However, the learner is allowed to request labels using a more powerful requester class than $\mathcal{R}_\rho$ during training. As in classical PAC-learning results, improper learning allows the circumvention of the impossibility result.

As a practical matter, the improper setting is useful when the system designer is willing to spend more resources during training. In our motivating example, the designer of the abuse-detection system might be willing to implement a more complicated system during training, which might include a larger budget (in dollars or man-power) for human intervention. However, after a time horizon $T$, training stops, and the designer commits to some $h_T, r_T$ for $r_T \in \mathcal{R}_\rho$. From then on, examples satisfyng $r_T(x) = 0$ are routed to $h_T$, and thus, we wish to characterize $\mathcal{L}_{\mathcal{H}, \mathcal{R}_\rho}(h_T, r_T)$.

In the improper setting, we demonstrate that IWAL, an algorithm from the active learning literature [Beygelzimer et al., 2009], can be adapted to our setting into an algorithm which we call DPL-IWAL. Since our objective is no longer an expectation of some loss function, but rather, the conditional expectation evaluated on the event that $r(x) = 0$, IWAL's standard analysis does not apply. A key technical hurdle is proving that estimates of this conditional loss concentrate at the right rate in order to attain generalization guarantees for the pair $(h, r)$ returned by DPL-IWAL. We show that over a time horizon $T$, DPL-IWAL algorithm requests $O((\rho + \eta)T)$ examples where $\eta = \min_{(h,r) \in \mathcal{H} \times \mathcal{R}_\rho} \mathbb{E}[L(h(x), y) \mid r(x) = 0]$ is the optimum value of our objective. At the same time our main lower bound demonstrates that $\Omega((\rho + \eta)T)$ requests are in fact necessary to compete with the best policy in $\mathcal{H} \times \mathcal{R}_\rho$.

Finally, we conduct a thorough exploration of these techniques on a number of datasets. We first undertake an ablation study, using a finite hypothesis class, showing that DPL-IWAL outperforms baselines that either ignore the active learning or abstention learning aspects of the problem. DPL-IWAL is not computationally efficient to implement since just like IWAL, it maintains a version space, that is a set of candidate hypotheses, which is non-trivial to optimize over in general. We therefore also conduct a number of experiments with an efficient heuristic inspired by our results using continuous hypothesis classes, outperforming natural baselines including margin sampling [Lewis and Gale, 1994, Balcan et al., 2007], which admits state-of-the-art performance for active learning problems in practice [Yang and Loog, 2016, Mussmann and Liang, 2018, Chuang et al., 2019].

## 1.1    Related Work

Our setting encompasses an objective considered in abstention learning (sometimes called selective classification), where the learner controls its evaluation region, that is the region of the domain where the learner is evaluated, by abstaining on the complement. In our setting, the evaluation region is where the learner does not request a label and the complementing abstention region is where the learner makes a request. At the same time, our setting considers the feedback mechanism from active learning. These are tied in a specific way in our framework: feedback is only available on the abstention (i.e. requesting) region and no information is revealed about the evaluation region. Neither an abstention algorithm nor a standard active learning algorithm would work in our setting and the algorithms in these settings can lead to incorrect solutions (see Appendix C). Nevertheless, we survey some of the relevant literature.

Learning with abstention was first studied by Chow [1957, 1970] for specific practical applications. Subsequently, several authors analyzed algorithms [Bartlett and Wegkamp, 2008, Grandvalet et al., 2008, Yuan and Wegkamp, 2010, Yuang and Wegkamp, 2011] for this setting with an emphasis on developing margin-based rules for abstention. Along a different line of work, El-Yaniv and Wiener [2010, 2011] analyze the theoretical trade-off between the coverage of an abstention function and a classifier's performance when not abstaining. All these works share in common the assumption that the learner has offline access to fully labeled samples, unlike the setting considered in this work.

A more recent line of work Cortes et al. [2016a,b, 2018] considers a setting where the algorithms learn over two classes of functions, a hypothesis class and an abstention class. These work either assume full feedback is always available (e.g. Cortes et al. [2016a,b]), or like in Cortes et al. [2018], the feedback is only available in the evaluation region, which is the exact opposite feedback mechanism than that of our setting since we receive feedback in the abstention region. Similarly, Shekhar et al. [2021] consider a setting where feedback is always available.

In the active learning literature, several authors focus on analyzing margin-based active learning which requests the labels for points close to a learned model classification surface [Dasgupta et al., 2005, Balcan et al., 2007, Balcan and Long, 2013, Awasthi et al., 2014, 2015, Zhang, 2018, Huang et al., 2019, Zhang et al., 2020]. Other algorithms admit generalization guarantees on the same order as passive learning while proving that their algorithm's label complexity, i.e. the number of points requested during learning, is bounded by a favorable rate. Beygelzimer et al. [2009] derived an algorithm, called IWAL, for general loss functions with strong theoretical guarantees. We wish to use this algorithm for our setting with the abstention learning objective. However, the loss function we consider is in fact a conditional expectation evaluated on an event, which is not amenable to standard IWAL. Our analysis and corresponding algorithm, DPL-IWAL, describes how to apply IWAL to such a setting.

Finally, ideas from the abstention framework have been applied to the active learning previously. Zhang and Chaudhuri [2014] used confidence-based predictors as a subroutine of an active learning algorithm. El-Yaniv and Wiener [2012] applied an abstention strategy from El-Yaniv and Wiener [2010] to the CAL algorithm of Cohn et al. [1994] proving theoretical guarantees, but only under specific model and distributional assumptions, which we do not make in our setting.

## 2  Setting and Preliminaries

In the dual purpose labeling framework, a learner is given a hypothesis class $\mathcal{H}$ with finite VC-dimension $d$, and a class of deterministic requester functions $\mathcal{R} \subset \{X \to \{0,1\}\}$. Nature fixes a distribution $\mathcal{D}$ over $\mathcal{X} \times \mathcal{Y}$, unknown to the learner. Given the marginal over $\mathcal{X}$, and $\rho > 0$, we denote $\mathcal{R}_\rho \subset \mathcal{R}$ as the subset of $\mathcal{R}$ with bounded request-rate $\mathbb{E}_x[r(x) = 1] \leq \rho$ for all $r \in \mathcal{R}_\rho$. $\mathbb{E}_x[r(x) = 1]$ can be well estimated using unlabeled data, which is generally readily available, allowing a bound on request-rate to be enforced in practice.

The interaction between the learner and nature proceeds through a sequence of rounds $t$. The learner first selects $(h_t, r_t) \in \mathcal{H} \times \mathcal{R}_\rho$ as a function of the past. Nature then draws an independent sample $(x_t, y_t)$ from $\mathcal{D}$, where $x_t$ is revealed to the learner. If $r_t(x_t) = 1$, the learner additionally observes $y_t$, but the performance of $h_t$ is not evaluated. If $r_t(x_t) = 0$, the learner does not observe $y_t$, but the performance of $h_t$ is evaluated. Thus, the choice of $r_t$ serves dual purposes: it determines whether the algorithm receives feedback, and whether its performance will be evaluated.

Formalizing this further, we suppose that the learner is given a loss function $L : \mathcal{Y} \times \mathcal{Y} \to \mathbb{R}$. The learner seeks to output $h_t$ that generalizes well on the region where $r_t$ dictates that $h_t$ should be evaluated. We therefore seek to bound the (conditional) excess loss $\mathcal{L}_{\mathcal{H}, \mathcal{R}_\rho}(h_t, r_t) = E[L(h_t(x), y) \mid r_t(x) = 0] - \inf_{(h^*, r^*) \in \mathcal{H} \times \mathcal{R}_\rho} \mathbb{E}[L(h^*(x), y) \mid r^*(x) = 0]$. In particular we seek $O(1/\sqrt{t})$ bounds on $\mathcal{L}_{\mathcal{H}, \mathcal{R}_\rho}(h_t, r_t)$, matching the generalization rate of full-feedback passive learning.

The conditional loss of the best pair in $\mathcal{H} \times \mathcal{R}_\rho$ plays an important role in our lower bounds, and so it is useful to define $\eta = \inf_{(h^*, r^*) \in \mathcal{H} \times \mathcal{R}_\rho} E_{x,y}[L(h^*(x), y) \mid r^*(x) = 0]$.

Finally, we call this the *proper dual purpose framework* since the algorithm is attempting to generalize well with respect to the class $\mathcal{H} \times \mathcal{R}_\rho$ and labels are generated according to functions in $\mathcal{R}_\rho$. In the

subsequent section, we will see that $O(1/\sqrt{t})$ bounds on $\mathcal{L}_{\mathcal{H},\mathcal{R}_\rho}(h_t, r_t)$ are impossible in general. This motivates the *improper dual purpose framework* as introduced in the following section.

## 3 Lower Bound

In this section, we present a lower bound stating that it is impossible for any algorithm to achieve an $O(1/\sqrt{t})$ generalization rate in the proper dual purpose setting. Thus, we here introduce the notion of *improper algorithms*. On each round $t$, an improper algorithm selects $h_t \in \mathcal{H}$ and an $R_t : X \to \{0, 1\}$ that is unconstrained for the purposes of showing a lower bound (e.g. $R_t$ is not necessarily in the set $\mathcal{R}_\rho$). As in the proper setting, label requests are tied to the evaluation region. That is, the algorithm sees $y_t$ i.f.f. $R_t(x_t) = 1$, and wishes to minimize $\mathcal{L}_{\mathcal{H},\mathcal{R}_\rho}(h_t, R_t)$. Notice that $\mathcal{L}$ is still defined with respect to the reference class $\mathcal{R}_\rho$, and that the proper setting is a special case of the improper setting where $R_t$ is equal to an $r_t \in \mathcal{R}_\rho$. The lower bound below shows that any algorithm with the desired generalization rate must satisfy $\frac{1}{T}\sum_{t=1}^{T} E[R_t(x_t)] > \rho$. Thus, since $E[r(x)] \le \rho$ for every $r \in \mathcal{R}_\rho$, no proper algorithm can attain the desired rate.

We prove a bound that holds for almost any possible classes of functions $\mathcal{H}, \mathcal{R}$, with some restrictions on $\mathcal{R}$. We say that $\mathcal{R}$ separates $\mathcal{X}$ if given any finite set of examples in $x_0, \dots, x_n \in \mathcal{X}$, there exists a point $\hat{x}$ outside this finite set of points and a requester in $\mathcal{R}$ such $r(\hat{x}) = 1$ while $r(x_0) = \dots = r(x_n) = 0$. This condition is much stronger than is necessary for the lower bound, but is already satisfied by simple classes such as when $\mathcal{R}$ contains linear separators and $\mathcal{X}$ is a ball in any dimension $\ge 2$, and becomes easier to satisfy as $\mathcal{R}$ becomes more complex. We need a weaker condition that requires only the separation property hold for *some* set shattered by $\mathcal{H}$.

**Definition 1.** $\mathcal{R}$ *separates* $\mathcal{X}$ *with respect to* $\mathcal{H}$ *if* $d = \mathrm{VCD}(\mathcal{H})$, *and there exists a set of* $d$ *examples* $x_0, \dots, x_{d-1}$ *shattered by* $\mathcal{H}$, $\hat{x} \in \mathcal{X}$, *and* $r \in \mathcal{R}$ *such that* $r(\hat{x}) = 1$ *and* $r(x_0) = \dots r(x_{d-1}) = 0$.

We first describe a distribution that follows the basic construction used to demonstrate lower bounds for pure active learning [Beygelzimer et al., 2009]. We then augment this distribution to include a region of mass $\rho$ that contains random noise. Intuitively, one can think of an algorithm as requesting a label either to solve the active learning problem or to avoid loss on examples that it is uncertain on in the region of mass $\rho$. We argue that the optimal algorithm, when not solving the active learning problem, spends $\rho T$ labels requesting on the random noise.

While this is the basic idea, the proof needs to preclude the possibility that an algorithm with a suboptimal prediction $h$ benefits from requesting labels outside the region of mass $\rho$ purely for the purpose of avoiding loss (and not to solve the active learning problem). An algorithm can also request labels at a rate greater than $\rho$ on any given round with the hope of decreasing its overall label complexity over $T$ rounds. The proof shows that neither of these strategies benefits an algorithm enough to deviate from the optimal strategy outlined in the previous paragraph.

**Definition 2.** *Fix any* $\rho \ge 0$. *We say that a round* $t$ *is a* failed round *if the algorithm selects a requesting strategy* $R_t : \mathcal{X} \to \{0, 1\}$, *and hypothesis* $h_t \in \mathcal{H}$ *satisfying* $\mathbb{E}[L(h_t(x), y) \mid R_t(x) = 0] \ge \min_{(h,r) \in \mathcal{H} \times \mathcal{R}_\rho} \mathbb{E}[L(h(x), y) \mid r(x) = 0] + \sqrt{\frac{d\eta}{t}}$.

As a practical matter, we will also require that an improper algorithm outputs a model $(h_T, r_T) \in \mathcal{H} \times \mathcal{R}_\rho$ at the end of its time horizon, where $r_T$ comes from the reference class. This paves the way for algorithms discussed in the subsequent sections which are applicable in systems that are able to tolerate a higher labeling overhead during a finite training horizon, but eventually need to converge on a model with request rate $\rho$. Crucially, the lower bound establishes the minimum additional overhead during training as $\eta$ (defined in Section 2), which is matched by our upper bounds.

**Theorem 1.** *Let* $L(h(x), y) = 1[yh(x) \le 0]$ *be the misclassification loss. Given* $\mathcal{R}$ *that* separates $\mathcal{X}$ *with respect to* $\mathcal{H}$ *with* $d = \mathrm{VCD}(\mathcal{H})$, *let* $\mathcal{R}_\rho \subset \mathcal{R}$ *consist of requesters with bounded request-rate* $\rho$. *For any* $\eta \le 1/4$, $\rho \le 1/2$, *there exists a distribution on* $\mathcal{X} \times \{-1, 1\}$ *such that* $\eta = \min_{(h,r) \in \mathcal{H} \times \mathcal{R}_\rho} \mathbb{E}[L(h(x), y) \mid r(x) = 0]$.

*Furthermore, there exists a sufficiently large* $T \ge 0$ *such that with probability at least* $1/2$ *any algorithm that, (A) outputs* $(h_T, r_T) \in \mathcal{H} \times \mathcal{R}_\rho$, *such that* $\mathbb{E}[L(h_T(x), y) \mid r_T(x) = 0] \le \eta + \sqrt{d\eta/T}$, *and (B) suffers no more than* $T/2$ *failed rounds, requires that:* $\mathbb{E}[\sum_t R_t] \ge \Omega((\eta + \rho)T)$.

**Algorithm 1** DPL-IWAL Algorithm

---

**Require:** Max iteration $T > 0$, $V_1 = \mathcal{H} \times \mathcal{R}_\rho$, $t_0$ be the first time $t$ such that $t \geq 16 \log(t/\delta)$
  **for** $t \in [1, t_0]$ **do**
    Observe $x_t$ in order to construct estimates $\frac{1}{t-1} \sum_{s=1}^{t-1} \mathbf{1}[r(x_s) = 0]$
  **for** $t \in [t_0 + 1, T]$ **do**
    $(h_t, r_t) \leftarrow \operatorname{argmin}_{(h,r) \in V_t} \widehat{L}_{t-1}(h, r)$
    Receive $x_t$
    **if** $r_t(x_t) = 1$ **then** Request label $y_t$
    $p_t(x_t) \leftarrow \min\left(1, \max_{(h,r),(h',r') \in V_t} \max_{y \in Y} \left| \frac{L(h(x_t),y)\mathbf{1}[r(x_t)=0]}{\frac{1}{t-1}\sum_{s=1}^{t-1}\mathbf{1}[r(x_s)=0]} - \frac{L(h'(x_t),y)\mathbf{1}[r'(x_t)=0]}{\frac{1}{t-1}\sum_{s=1}^{t-1}\mathbf{1}[r'(x_s)=0]} \right| \right)$
    $q_t \sim \text{Bernoulli}(p_t)$
    **if** $q_t = 1$ **then** Request or re-use label $y_t$
    $V_{t+1} \leftarrow \{(h,r) \in V_t : \widehat{L}_t(h,r) \leq \min_{(h,r) \in V_t} \widehat{L}_t(h,r) + \tilde{\Delta}_t\}$
    **if** $r_t(x_t) = 0 \wedge q_t = 0$ **then** Predict label using $\operatorname{sgn}(h_t(x_t))$
  **Return:** $(h_T, r_T)$

---

The above theorem states that an algorithm must request the labels of at least $\Omega((\eta + \rho)T)$ examples (in expectation) if we require that the pair returned by the algorithm generalizes at a rate approximating that of standard supervised learning. This then directly implies that $R_t$ must be selected outside of the class $\mathcal{R}_\rho$ since this class only contains functions with a requesting rate of at most $\rho$, resulting in label complexity at most $O(\rho T)$ in expectation. All proofs can be found in Appendix A.

Although we state the lower bound for classification loss, it can be extended to any loss function where mispredicting the sign of an example, $yh(x) \leq 0$, implies $L(h(x), y) \geq C$ for some constant $C$. This is true for the logistic, hinge, squared, and absolute losses.

## 4 Dual Purpose Labeling Algorithm

In this section, we present our algorithm, DPL-IWAL (see Algorithm 1 for the pseudo-code), in the improper dual purpose framework. At a high level, DPL-IWAL finds the pair $(h, r) \in \mathcal{H} \times \mathcal{R}_\rho$ that minimizes the conditional loss, $\mathbb{E}[L(h(x), y)|r(x) = 0]$, i.e. the expected loss of $h$ conditioned on the event that the label is not requested by $r$. Intuitively, the best pair $(h, r)$ requests the label of the point whenever the prediction of $\operatorname{sgn}(h(x))$ is likely to be incorrect.

To find such a pair, at each round $t$, the algorithm first constructs an importance weighted estimate, $\widehat{L}_t(h, r)$, of $\mathbb{E}[L(h(x), y)|r(x) = 0]$ by using the fewest number of labeled points as possible and then chooses the pair $(h_t, r_t)$ that minimizes $\widehat{L}_t(h, r)$. Ideally, the importance weighted estimates $\widehat{L}_t(h, r)$ could be constructed from the set of points whose labels have been requested by $r_1, \ldots, r_t$, but these sets of points are a non-trivially biased sample of the underlying distribution. Moreover, these points could reside in regions of the space that are not useful for calculating $\mathbb{E}[L(h(x), y)|r(x) = 0]$. To see this more clearly, consider a simple case when $\mathcal{R}_\rho$ is contains just one function and the algorithm is then simply finding the $\operatorname{argmin}_{h \in \mathcal{H}} \mathbb{E}[L(h(x), y)|r(x) = 0]$. The points the $r$ functions request the label for, meaning points where $r(x) = 1$, do not reveal any information necessary to estimate $\mathbb{E}[L(h(x), y)|r(x) = 0]$ for $h \in \mathcal{H}$. Thus, the algorithm must label other regions in the space. Below, we describe how the algorithm uses a subset of the points requested by the $r_t \in \mathcal{R}_\rho$ in conjunction with some additional carefully chosen points, via a function $q_t$ outside $\mathcal{R}_\rho$, to construct unbiased estimators, $\widehat{L}_t(h, r)$. This fact thus makes DPL-IWAL an improper dual purpose algorithm.

Similarly to IWAL [Beygelzimer et al., 2009], the DPL-IWAL algorithm constructs an importance weighted estimate, but instead of estimating the expected loss as is done in IWAL, we craft an estimate of the conditional losses for $t > t_0$,

$$\widehat{L}_t(h, r) = \frac{1}{t - t_0 - 1} \sum_{s=t_0+1}^{t} \frac{q_s}{p_s(x_s)} \frac{L(h(x_s), y_s)\mathbf{1}[r(x_s) = 0]}{\frac{1}{s-1}\sum_{s'=1}^{s-1} \mathbf{1}[r(x_{s'}) = 0]},$$

where a coin $q_s \in \{0, 1\}$ is flipped with a bias probability $p_s(x_s)$ and where $t_0$ is the first time $t$ such that $t \geq 16 \log(t/\delta)$. Note that for the first $s \in [1, t_0]$, we simply observe the features

$x_s$ in order to construct $\frac{1}{s-1}\sum_{s'=1}^{s-1}\mathbf{1}[r(x_{s'})=0]$ that are non-zero with high probability since these are needed in definition of the denominator of $\widehat{L}_t(h,r)$. Ignoring the $q_s$ and $p_s(x_s)$ for now, the numerator $\frac{1}{t-t_0-1}\sum_{s=t_0+1}^{t}L(h(x_s),y_s)\mathbf{1}[r(x_s)=0]$ is a measure of the joint expectation $\mathbb{E}[L(h(x),y)\mathbf{1}[r(x)=0]]$ while the denominator contains running averages of the $\mathbb{E}[r(x)=0]$. Roughly speaking, by considering the ratio of these two terms, we estimate the conditional expected loss, $\mathbb{E}[L(h(x),y)|r(x)=0]=\frac{\mathbb{E}[L(h(x),y)\mathbf{1}[r(x)=0]]}{\mathbb{E}[r(x)=0]}$.

The algorithm maintains a version space, $V_t$, as defined in Algorithm 1, which it reduces at each round. We prove that it suffices to use a slack term $\tilde{\Delta}_t=\tilde{O}\big(\sqrt{(1/t)\log(1/\delta)}\big)$, in order to ensure with high probability that $(h^*,r^*)$ remain within the version space as it shrinks. The $\tilde{O}(\cdot)$ hides constants and $\log(t|\mathcal{H}\times\mathcal{R}_\rho|)$ factors; see the appendix for exact constants. In order to reduce the number of labeled points used to construct the importance-weighted estimates, the probability of requesting a point $p_t$ is defined by the (estimated) conditional loss difference between pairs of functions in this shrinking set $V_t$. Given the above, the algorithm's overall requesting rule, $R_t$, is thus defined by the following condition: $R_t(x_t)=1$ if and only if $r_t(x_t)=1\vee q_t=1$ where $r_t\in\mathcal{R}_\rho$ and $q_t\notin\mathcal{R}_\rho$.

# 5   Generalization and Label Complexity Guarantees

In this section, we present a series of theoretical guarantees that analyze the performance of our approach as compared to different baselines as well as prove an upper bound on the expected number of label requests by the DPL-IWAL algorithm.

In our framework, we seek to select a hypothesis $h_t$ which incurs minimal loss whenever a label request is not made. Below, we prove an upper bound on this type of loss that is in terms of the best pair of functions, $(h^*,r^*)=\operatorname{argmin}_{(h,r)\in\mathcal{H}\times\mathcal{R}_\rho}\mathbb{E}[L(h(x),y)|r(x)=0]$.

**Theorem 2.** *Given any $\rho<\frac{1}{2}$, for any $\delta>0$, with probability at least $1-\delta$, for all $t\geq 16\log(3t/\delta)$,*
$$\mathbb{E}[L(h_t(x),y)|r_t(x)=0]\leq\mathbb{E}[L(h^*(x),y)|r^*(x)=0]+\tilde{O}\big(\sqrt{(1/t)\log(1/\delta)}\big).$$

This guarantee states that the pair $(h_t,r_t)$ chosen by the algorithm is converging to the best pair $(h^*,r^*)$ as a function of the time $t$ with respect to the conditional loss. The assumption $t\geq 16\log(3t/\delta)$ is mild condition, for example, if $\delta=0.0001$ we then require $t>104$. Also, in most standard applications, only a small fraction of examples can be labeled and it is natural to assume that $\rho<\frac{1}{2}$. Nevertheless, this assumption can be reduced by increasing the constraint on $t$ in the bound.

Overall, the theoretical analysis, which is given in Appendix B, departs from standard derivations since we need to carefully deal with conditional losses and the constructed estimates, $\widehat{L}_t(h,r)$. More concretely, we first must ensure that the denominator of the estimate $\widehat{L}_t(h,r)$ is non-zero with high probability as otherwise the estimate would not be well defined. To do so, we start the labeling process of our algorithm only after $t_0$ examples have been observed. After $t_0$ examples have been observed and using the fact that $\mathbb{E}[r(x)=0]>1-\rho$, we can then prove that the condition $\frac{1}{s-1}\sum_{s'=1}^{s-1}\mathbf{1}[r(x_{s'})=0]>0$ holds with high probability (Lemma 1). Then, in order to apply Azuma's inequality on conditional losses, we need to prove that $\frac{L(h(x_s),y_s)\mathbf{1}[r(x_s)=0]}{\frac{1}{s-1}\sum_{s'=1}^{s-1}\mathbf{1}[r(x_{s'})=0]}$ is bounded by a favorable constant despite the variable denominator term (Lemma 2). Azuma's inequality implies that the estimates are converging to $\mathbb{E}[\widehat{L}_t(h,r)]$, but this is not enough since we want to prove guarantees in terms of $\mathbb{E}[L(h(x),y)|r(x)=0]$. Thus, using a series of concentration inequalities, we prove that expected value of the estimate, $\mathbb{E}[\widehat{L}_t(h,r)]$, converges to the expected conditional loss, $\mathbb{E}[L(h(x),y)|r(x)=0]$ at the desired rate (see proof of Theorem 2).

Next, we compare the quality of the predictions of our approach to that of simply predicting according to the best-in-class as measured by the non-conditional loss, $h_b=\operatorname{argmin}_{h\in\mathcal{H}}\mathbb{E}[L(h(x),y)]$. This comparison quantifies the potential benefits of our framework as compared to that of supervised learning since $h_b$ is the hypothesis that an algorithm in the supervised setting is attempting to learn.

In the next corollary, we consider the never-requester function, $r_\diamond$, that is $\mathbf{1}[r_\diamond(x)=0]=1$ for all $x\in\mathcal{X}$. The never-requester is in $\mathcal{R}_\rho$ for any value of $\rho>0$. The never-requester trivially does not increase the label complexity and since it's a single function, it also does not discernibly augment the complexity of class, $\mathcal{R}_\rho$, and so it can be included in $\mathcal{R}_\rho$ at effectively no cost.

**Corollary 1.** *Given any* $\rho < \frac{1}{2}$, *for any* $\delta > 0$, *with probability at least* $1 - \delta$, *for all* $t \geq 16 \log(3t/\delta)$, $\mathbb{E}[L(h_t(x), y)|r_t(x) = 0] \leq \mathbb{E}[L(h_b(x), y)] + \gamma + \tilde{O}\big(\sqrt{(1/t)\log(1/\delta)}\big)$, *where* $\gamma = \mathbb{E}[L(h^*(x), y)|r^*(x) = 0] - \mathbb{E}[L(h_b(x), y)]$. *Furthermore, if* $r_\diamond \in \mathcal{R}_\rho$, *then* $\gamma \leq 0$.

The above corollary states the predictions of the chosen function $h_t$ when not requesting admit strictly fewer mistakes as compared to the predictions of $h_b$, the best-in-class in $\mathcal{H}$, whenever $\gamma < -\tilde{O}\big(\sqrt{(1/t)\log(1/\delta)}\big)$. The value $\gamma$ characterizes the difference between the best-in-class in our setting versus the best-in-class in the supervised learning. The more negative this term is, the fewer number of mistakes are made. In an empirical study in Appendix D, we show that typically $\gamma$ is significantly smaller than 0.

To derive label complexity guarantees, we define a disagreement coefficient for conditional losses, directly derived from the coefficient definitions in Henneke [2007], Beygelzimer et al. [2009]. Let $\rho((h, r), (h', r')) = \mathbb{E}[|\frac{L(h(x), y)\mathbf{1}[r(x) = 0]}{\mathbb{E}[r(x) = 0]} - \frac{L(h'(x), y)\mathbf{1}[r'(x) = 0]}{\mathbb{E}[r'(x) = 0]}|]$ be a measure of the distance between two pairs $(h, r)$ and $(h', r')$. Based on this metric, we define the ball around the best pair $(h^*, r^*)$ as follows: $B(h^*, r^*, \Lambda) = \{(h, r) \in \mathcal{H} \times \mathcal{R}_\rho : \rho((h, r), (h^*, r^*)) \leq \Lambda\}$. The disagreement coefficient is the infimum value of $\theta > 0$ such that for all $\Lambda \geq 0$: $\mathbb{E}\left[\max_{(h, r) \in B(h^*, r^*, \Lambda)} \max_y \left|\frac{L(h(x), y)\mathbf{1}[r(x) = 0]}{\mathbb{E}[r(x) = 0]} - \frac{L(h^*(x), y)\mathbf{1}[r^*(x) = 0]}{\mathbb{E}[r^*(x) = 0]}\right|\right] \leq \theta\Lambda$. The next theorem bounds the expected number of points requested needed to construct the estimates, $\hat{L}_t(h, r)$ in terms of the coefficient $\theta$.

**Theorem 3.** *Given any* $\rho < \frac{1}{2}$, *for all* $\delta > 0$, *with probability at least* $1 - \delta$, $\sum_{s=1}^T \mathbb{E}[p_s(x_s)] = \tilde{O}(\theta\eta T + \theta\sqrt{T})$, *where* $\theta$ *is the disagreement coefficient.*

Since the labeling rate of $r_t$ is at most $\rho$, the label complexity of the DPL-IWAL algorithm is then given by $\sum_{s=1}^T \mathbb{E}[r_s(x_s) = 1] + \mathbb{E}[p_s(x_s)] = \tilde{O}((\theta\eta + \rho)T + \theta\sqrt{T})$. Assume that $\mathbb{E}[L(h_t(x), y) \mid R_t(x) = 0] \leq \mathbb{E}[L(h_t(x), y)|r_t(x) = 0]$; intuitively, this implies using $q_t$ in addition to $r_t$ to make requests helps reduce our conditional loss and it holds in practice as shown by our experiments in Appendix D. Then it follows, by Theorem 2, that $\mathbb{E}[L(h_t(x), y) \mid R_t(x) = 0] \leq \eta + \tilde{O}(\sqrt{(1/t)\log(1/\delta)})$, i.e. with high probability failed rounds do not occur. Thus, in this case, DPL-IWAL exhibits an upper bound on the label complexity that matches the lower bound stated in the previous section, apart for $o(T)$ terms, namely $\sqrt{T}$. An even sharper bound for the $o(T)$ term is possible, using analysis similar to that of the EIWAL algorithm in Cortes et al. [2019], resulting in rate of $\sqrt{\eta T}$.

The analysis that proves the label complexity bound carefully deals with the denominator term of the estimates of the conditional loss as well as the fact that we are working with two function classes. Similarly to the generalization bound analysis, we first prove that the denominator term is well behaved and is not too far form its mean. Then, we leverage the disagreement coefficient $\theta$ and shrinking version space over the two classes.

In this paper, we analyzed the scenario where the requesting functions $r_t \in \mathcal{R}_\rho$ are constrained to request at most $\rho$ times and minimize the conditional loss over this set of requesters. One could instead consider a Lagrangian relaxation of this constraint and pay a fixed cost $c$ for each request. That is, consider the cost-based loss defined by $\mathbf{1}_{r(x)=0}L(h(x), y) + c\mathbf{1}_{r(x)=1}$. Both of these loss views are important and in fact have been analyzed in the abstention setting (e.g., see El-Yaniv and Wiener [2010] for conditional loss and see Cortes et al. [2016a] for cost-based loss). Despite focusing on conditional loss over constrained requester functions, the theory and algorithms in this paper can be extended to the cost-based loss. In particular, for the algorithm not to incur too many mistakes during training and to return a pair of function that generalized well, the algorithm must select $R_t$ outside of the class of requesters. The version of DPL-IWAL for cost-based loss will thus require requesting according to both $q_t$ and $r_t$.

## 6  Empirical Investigation

We start our empirical investigation by corroborating the theoretical insights made in the previous sections with an ablation study of DPL-IWAL. Since IWAL needs to solve a computationally intractable constrained optimization problem over its version space, which is only made more challenging in our setting by the joint optimization over $\mathcal{H} \times \mathcal{R}_\rho$, we use finite classes for these initial

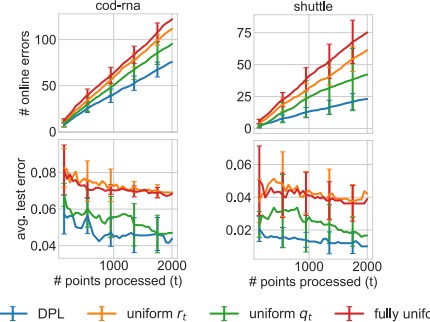

**DPL-Simplified Algorithm**

**Require:** Max iteration $T > 0$, classes $\mathcal{H} \times \mathcal{R}_\rho$
  **for** $t \in [1, T]$ **do**
    $h_t \leftarrow \mathrm{argmin}_{h \in \mathcal{H}} \sum_{s=1}^{t-1} q_s L(h(x_s), y_s)$
    $r_t \leftarrow \mathrm{argmin}_{r \in \mathcal{R}_\rho} \sum_{s=1}^{t-1} q_s \frac{L(h_t(x_s), y_s)\mathbf{1}[r(x_s)=0]}{\frac{1}{s-1}\sum_{s'=1}^{s-1}\mathbf{1}[r(x_{s'})=0]}$
    Receive $x_t$
    **if** $r_t(x_t) = 1$ **then** $q_t \leftarrow 1$, request label $y_t$
    **else** $q_t \leftarrow 0$, predict label using $\mathrm{sgn}(h_t(x_t))$
  **Return:** $(h_T, r_T)$

Figure 1: On the left, the number of online mistakes made while processing a stream of data and the held-out conditional loss on non-requested points made by DPL-IWAL and baselines comparators. The plots show the mean and standard deviation over 10 trials. On the right, the pseudo-code of DPL-Simplified Algorithm.

experiments. Then, we turn to a more practical algorithm inspired by the DPL-IWAL algorithm, but that leverages readily available optimization routines and empirically effective heuristics.

**Ablation Study:** We compare the performance of DPL-IWAL against baselines which ignore either the active learning and/or the abstention aspects of the dual purpose labeling, in order to demonstrate that indeed both aspects are necessary. To do so, we compare our algorithm to several variants, defined as follows. The *fully uniform* baseline simply decides to request a label using a random biased coin-flip independent of the example, thereby mimicking standard passive supervised learning. The *uniform $r_t$* baseline uses the DPL-IWAL algorithm in conjunction with trivial requester class that fixes its output uniformly at random, independent of the input $x_t$. The *uniform $q_t$* baseline is similar to DPL-IWAL, with outcomes $q_t$ determined by coin-flips with a fixed bias, independent of $x_t$.

For this experiment, we define a set of requester regions near the margin of the classification surface, each covering a mass $\rho$ of the overall distribution. Specifically, for any set of real-valued hypotheses set $\mathcal{H}$, where the classification of point $x$ made by $h \in \mathcal{H}$ is defined as $\mathrm{sgn}(h(x))$, we define a margin-based requester function class as: $\mathcal{R}_{\rho, \mathcal{H}} = \{r_h(x) \mapsto \mathbf{1}[|h(x)| \leq \tau_{h,\rho}(\mathcal{D}_\mathcal{X})] : h \in \mathcal{H}\}$, where $\mathcal{D}_\mathcal{X}$ is marginal distribution on $\mathcal{X}$ and $\tau_{h,\rho}(\mathcal{D}_\mathcal{X})$ is the largest threshold value that satisfies $\mathbb{E}_{\mathcal{D}_\mathcal{X}}[r_h(x) = 1] \leq \rho$. Note, the $\tau_{h,\rho}$ threshold can be estimated using unlabeled data.

We test six publicly available datasets [Chang and Lin] and for each, we use linear logistic regression models trained using the Python `scikit-learn` library. For all datasets, we construct a finite hypothesis class $\mathcal{H}$ and a matching finite margin-based requester set $\mathcal{R}_{\rho, \mathcal{H}}$. We then stream unlabeled examples to each algorithm by sampling without replacement from a training split. The process is repeated 10 times, using a different random train/test split for each trial. For details, see Appendix D.

In Figure 1 (left), we compare each method using two metrics. First, we consider the number of incorrect predictions (i.e. "online mistakes") that the model makes while processing the stream of unlabeled training data, where label-requests are spared mistakes. Second, we consider the conditional loss of the currently selected pair $(h_t, r_t)$ by measuring the average misclassification loss on non-requested points using a held-out test split. See Appendix D for our results on all datasets, which all show a similar pattern.

Overall these results show that our DPL-IWAL algorithm, with both a non-trivial sampling function $q_t$ and requesting function $r_t$, outperforms all baseline methods. This indicates that both active learning and abstention aspects of DPL-IWAL are necessary since naively applying an active learning algorithm without abstention (e.g. uniform $r_t$) or naively applying an abstention algorithm without active learning (e.g. uniform $q_t$) admit suboptimal results.

**DPL-Simplified Algorithm:** Here, we consider the DPL-Simplified Algorithm (Figure 1 right) where, the joint optimization problem over $\mathcal{H} \times R_\rho$ is split into two separate optimizations. This piece-wise optimization may not arrive at the same solution as the joint optimization, but we find it to be an empirically effective proxy. The first optimization over $\mathcal{H}$ is a standard learning problem over the currently labeled examples and any off-the-shelf hypothesis class and training algorithm can be used. The second optimization over $\mathcal{R}_\rho$ is still non-trivial to solve. However, the objective

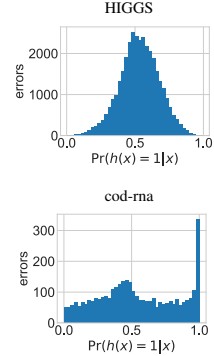

| dataset | # Online Mistakes | | | Conditional Test Error (%) | | |
|---|---|---|---|---|---|---|
| | KNN | MAR | MIX | KNN | MAR | MIX |
| a9a | $2017 \pm 35$ | $\mathbf{1531 \pm 44}$ | $1713 \pm 51$ | $12.1 \pm 0.39$ | $\mathbf{8.9 \pm 0.44}$ | $10.3 \pm 0.30$ |
| acoustic | $1654 \pm 58$ | $\mathbf{1458 \pm 52}$ | $1418 \pm 51$ | $9.4 \pm 0.35$ | $8.1 \pm 0.13$ | $\mathbf{7.8 \pm 0.28}$ |
| cifar | $904 \pm 30$ | $849 \pm 38$ | $\mathbf{701 \pm 26}$ | $5.0 \pm 0.23$ | $5.1 \pm 0.14$ | $\mathbf{3.8 \pm 0.09}$ |
| cod-rna | $644 \pm 27$ | $360 \pm 13$ | $329 \pm 12$ | $3.1 \pm 0.11$ | $2.2 \pm 0.06$ | $\mathbf{1.5 \pm 0.07}$ |
| covtype | $3471 \pm 89$ | $3365 \pm 89$ | $3318 \pm 78$ | $18.6 \pm 0.30$ | $19.3 \pm 0.18$ | $\mathbf{17.9 \pm 0.28}$ |
| HIGGS | $5862 \pm 79$ | $5500 \pm 90$ | $5689 \pm 69$ | $35.2 \pm 0.31$ | $\mathbf{32.7 \pm 0.25}$ | $34.0 \pm 0.27$ |
| ijcnn | $654 \pm 52$ | $420 \pm 20$ | $310 \pm 30$ | $2.2 \pm 0.28$ | $2.4 \pm 0.16$ | $\mathbf{0.7 \pm 0.14}$ |
| mnist | $1489 \pm 69$ | $1032 \pm 26$ | $\mathbf{958 \pm 50}$ | $6.7 \pm 0.31$ | $5.1 \pm 0.09$ | $\mathbf{4.1 \pm 0.12}$ |
| shuttle | $\mathbf{29 \pm 16}$ | $122 \pm 27$ | $\mathbf{21 \pm 8}$ | $0.1 \pm 0.05$ | $0.8 \pm 0.09$ | $0.1 \pm 0.05$ |
| skin | $\mathbf{61 \pm 64}$ | $662 \pm 110$ | $\mathbf{36 \pm 31}$ | $0.1 \pm 0.02$ | $4.0 \pm 0.89$ | $\mathbf{0.02 \pm 0.01}$ |

Figure 2: The left side of the table displays the mean and standard deviation over 10 trials of the number of total online mistakes after processing 20,000 examples for each of the requestor strategies (all limited to requesting labels for 20% of examples). The right side of the table shows the mean and standard deviation of the conditional misclassification loss $\Pr(h(x) \neq y | r(x) = 0)$, measured on the test set, for $(h_T, r_T)$. On the right, histograms show the number of errors made by a partially-trained linear hypothesis $h$ as a function of the model confidence for two datasets.

does provide the intuition that an optimal requester seeks to cover all of classifier $h_t$'s mistakes. This suggests an approximate solution, where we train a requester $r$ that seeks to classify the incorrect predictions of a fixed hypothesis $h_t$ over the set of labeled examples thus far. At the same time, notice that the simplified algorithm fixes $q_t = r_t(x_t)$ (and no longer needs to solve IWAL's constrained optimization problem). This makes the requester function responsible for not just sampling regions of the space that $\mathcal{H}$ cannot correctly capture, but also sampling examples that are effective for training.

To cover the regions where the classifier is incorrect, we leverage what we call a **KNN-Requester** function. In particular, we use scikit-learn's KNeighborsClassifier to train a non-parametric model to predict hypothesis $h_t$'s training mistakes. The resulting requester is $r_{\text{KNN},h,\rho}(x) = \mathbf{1}[|1 - \Pr_\theta(h(x) \neq y|x)| < \tau_\theta]$, where $\Pr_\theta(h(x) \neq y|x)$ denotes the probability that $h$ makes a mistake according to the KNN model and $\rho$ indicates the classifier's threshold $\tau_\theta$ has been tuned so that the requester labels approximately a $\rho$ fraction examples. To select points that are effective for training the classifier, we borrow intuition from the simple yet empirically very effective *margin* (or *uncertainty*) active learning algorithm, which samples examples that the current model is least confident on (i.e., example closest to the decision surface).

This leads us to the **Mixture-Requester** (MIX) function which merges the margin and KNN strategies. Specifically, we uniformly combine the probability score produced by the KNN model underlying $r_{\text{KNN},h,\rho}$ and the margin score derived from $h_t(x)$ as follows: $r_{\text{MIX},h,\rho}(x) = \mathbf{1}[|1 - \Pr_\theta(h(x) \neq y|x)| + |\sigma(h(x)) - 0.5| < \tau_{h,\theta}]$, where $\sigma$ is a normalizing function that maps the input into $[0, 1]$, which in our case is the output of scikit-learn's predict_proba method. Thus, this requester seeks to sample points that are both covering mistakes of the classifier as well as sampling points that are effective for training.

In addition to the above, we evaluate the simpler **Margin-Requester** to serve as a natural, yet effective baseline: $r_{\text{MAR},h,\rho}(x) = \mathbf{1}[|\sigma(h(x)) - 0.5| < \tau_{h,\theta}]$. This requester is essentially mimicking the behavior of using uncertainty-based active learning without any regard to the DPL setting.

In the following experiments, the hypothesis class $\mathcal{H}$ is the set of linear models with bounded $L_2$-norm, trained using scikit-learn's LogisticRegression implementation, and we use 10 publicly available datasets, all which we cast as binary classification problems (see Appendix D for details). We execute a batch variant of DPL-Simplified, where at each iteration we process a batch of 5,000 examples, querying 20% of the examples for their labels and making prediction for the rest. Upon the completion of the iteration, we receive the requested labels and update the choice of $(h, r)$. This larger batch size, more closely reflects practical learning settings, where it is impractical to re-train the model after every single label query [Amin et al., 2020]. All methods are seeded with 500 randomly sampled initial examples and each experiment is run for 10 trials.

In Figure 2, we show both the number of online errors incurred during the iterative labeling/training procedure as well as the average conditional misclassification loss of the final $(h, r)$ pair on a test set.

In Appendix D, we plot the full learning curves in Figure 8 associated with Figure 2. Overall, MIX is very effective, outperforming the baseline methods in 8 out of 10 datasets. To better understand when MIX outperforms the margin-requester, we present a histogram of errors as a function the confidence of a model trained with a set of 3,500 examples sampled uniformly at random (Figure 2 right). The histogram for the HIGGS dataset, where the margin baseline performs well, shows that most of the errors are highly concentrated around the minimum model certainty region, i.e. where the model prediction score is close to $0.5$. In contrast, the histogram for cod-rna, where the DPL-inspired mixture-requester excels, shows that while there are some errors concentrated around the 0.5 threshold, there are also a number of errors far away from the model decision boundary.

## 7 Conclusion

We introduced a new setting which models the relationship between labeling and learning in real systems. We derived a lower bound in terms of the abstention-rate of a reference class and the optimum value of our objective. We presented an algorithm DPL-IWAL that admits strong generalization and label complexity guarantees and a more efficient variant, DPL-Simplified. Finally, we reported experiments which corroborate our theoretical findings and demonstrate that our algorithm outperforms natural baselines, including the ubiquitous margin sampling algorithm.

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
