# A Lower Bound Proof

In this appendix, we present the proof of the lower bound for the dual propose labeling problem analyzed in this paper.

**Theorem 1.** *Let $L(h(x), y) = 1[yh(x) \leq 0]$ be the misclassification loss. Given $\mathcal{R}$ that* separates $\mathcal{X}$ *with respect to* $\mathcal{H}$ *with* $d = \mathrm{VCD}(\mathcal{H})$*, let* $\mathcal{R}_\rho \subset \mathcal{R}$ *consist of requesters with bounded request-rate* $\rho$*. For any* $\eta \leq 1/4$*,* $\rho \leq 1/2$*, there exists a distribution on* $\mathcal{X} \times \{-1, 1\}$ *such that* $\eta = \min_{(h,r) \in \mathcal{H} \times \mathcal{R}_\rho} \mathbb{E}[L(h(x), y) \mid r(x) = 0]$*.*

*Furthermore, there exists a sufficiently large $T \geq 0$ such that with probability at least $1/2$ any algorithm that, (A) outputs $(h_T, r_T) \in \mathcal{H} \times \mathcal{R}_\rho$, such that $\mathbb{E}[L(h_T(x), y) \mid r_T(x) = 0] \leq \eta + \sqrt{d\eta/T}$, and (B) suffers no more than $T/2$ failed rounds, requires that: $\mathbb{E}[\sum_t R_t] \geq \Omega((\eta + \rho)T)$.*

*Proof.* Given an algorithm, define $Q = \mathbb{E}\left[\sum_t R_t\right]$, the expected number of labels requested by the algorithm. We begin by showing that under the conditions of the theorem, $Q = \Omega(\eta T)$. We use a similar construction as used for the lower bounds in standard active learning. However, we must be careful to ensure that the ability the learner has to abstain using $\mathcal{R}$ does not affect the bounds by too much. We then show that when $\rho > 2c\eta$, $Q = \Omega(\rho T)$, where $c$ is the constant in the definition of a failed round. Together, these two facts imply the theorem.

Let $x_0, x_1, \ldots, x_{d-1}$ and $\hat{x}$ be a set of examples satisfying Definition 1. Let $r_0 \in \mathcal{R}$ satisfy $r_0(\hat{x}) = 1$ and $r_0(x_i) = 0$, $i \geq 0$.

Fix an $\epsilon > 0$, and set $\beta = 2(\eta + 2\epsilon)$. Our marginal distribution on $\mathcal{X}$ will be supported on $\hat{x}, x_0, x_1, \ldots x_{d-1}$. The instance $\hat{x}$ will have probability mass $\rho$. $x_0$ will have probability mass $(1 - \rho)(1 - \beta)$. The remaining $x_i$ each have mass $(1 - \rho)\beta/(d - 1)$. Since $\mathbb{E}[r_0(x) = 1] = \mathbb{P}[x = \hat{x}] = \rho$, $r_0 \in \mathcal{R}_\rho$.

Let $\mathrm{Rad}(p)$ denote a Rademacher random variable with $p$ the probability of $+1$. Let $b_i \sim \mathrm{Rad}(1/2)$ for $i \in \{1, \ldots, d - 1\}$. These are determined once at the beginning of time.

For each round $t$ if $x_i$, $i \geq 1$, is selected by the marginal then $y$ is distributed as $\mathrm{Rad}(1/2 + \frac{\epsilon b_i}{\eta + 2\epsilon})$. If $x_0$ is selected by the marginal then $y = 1$ with probability 1. If $\hat{x}$ is selected by the marginal then $y$ is distributed as $\mathrm{Rad}(1/2)$.

For this distribution, the optimal hypothesis $h^*$ (regardless of which requester is used) labels $x_0$ as 1, labels each $x_i$, $i \geq 1$, as $b_i$ and labels $\hat{x}$ arbitrarily (say $h^*(\hat{x}) = 1$ wlog).

Intuitively, $r_0$ is the optimal requester in $\mathcal{R}_\rho$ for $h^*$ since, conditioned on $x$, $h^*$ suffers a loss of $1/2$ when $x = \hat{x}$, a loss of $(1/2 - \lambda)$ when $x = x_i, i \geq 1$ and a loss of 0 when $x = x_0$, where we define $\lambda = \frac{\epsilon}{\eta + 2\epsilon}$. (We make this fact precise below). However, for a *suboptimal* $h$, $h$ could in principle reduce its loss by carefully abstaining on examples it is uncertain on, avoiding examples where its conditional loss is $(1/2 + \lambda)$. In the following we bound the effectiveness of such a strategy.

Define $\mathcal{R}' = \{r : \mathcal{X} \to \{0, 1\} \mid r(\hat{x}) = 0\}$ as the set of all requesting strategies, not necessarily satisfying $\mathbb{P}[R'(x) = 0] \leq \rho$, and not necessarily belonging to $\mathcal{R}$, but that do not request a label for $\hat{x}$. $\mathcal{R}'_\rho = \{r \in \mathcal{R}' \mid \mathbb{P}[r(x) = 1] \leq \rho\}$.

Fix an arbitrary $h \in \mathcal{H}$, and $r \in \mathcal{R}'_\rho$. Let $F_i = \mathbf{1}[h(x_i) \neq b_i]$, and $G_i = 1 - F_i$. Define $Z = \mathbb{P}[r(x) = 1]$. Let $N = \frac{(1-\rho)\beta}{d-1}$ be the probability mass placed on each $x_i$, $i \geq 1$.

$$\mathbb{E}[L(h(x), y) \wedge r(x) = 0]$$
$$\geq \frac{\rho}{2} + N \sum_{i=1}^{d-1} \mathbf{1}[r(x_i) = 0]G_i(1/2 - \lambda) + N \sum_{i=1}^{d-1} \mathbf{1}[r(x_i) = 0]F_i(1/2 + \lambda)$$
$$= \frac{\rho}{2} + N \sum_{i=1}^{d-1} \mathbf{1}[r(x_i) = 0](1/2 - \lambda) + 2\lambda N \sum_{i=1}^{d-1} \mathbf{1}[r(x_i) = 0]F_i$$

$$= \frac{\rho}{2} + N \sum_{i=1}^{d-1}(1/2 - \lambda) + 2\lambda N \sum_{i=1}^{d-1} F_i$$

$$- N \sum_{i=1}^{d-1} \mathbf{1}[r(x_i) = 1](1/2 - \lambda) - 2\lambda N \sum_{i=1}^{d-1} \mathbf{1}[r(x_i) = 1]F_i$$

$$\geq \frac{\rho}{2} + N \sum_{i=1}^{d-1}(1/2 - \lambda) + 2\lambda N \sum_{i=1}^{d-1} F_i$$

$$- \frac{Z}{2} + \lambda N \sum_{i=1}^{d-1} \mathbf{1}[r(x_i) = 1] - 2\lambda N \sum_{i=1}^{d-1} \mathbf{1}[r(x_i) = 1]F_i$$

$$\geq \frac{\rho}{2} - \frac{Z}{2} + N \sum_{i=1}^{d-1}(1/2 - \lambda) + \lambda N \sum_{i=1}^{d-1} F_i$$

$$= \frac{\rho}{2} - \frac{Z}{2} + (1 - \rho)\beta(1/2 - \lambda) + (1 - \rho)\lambda\beta \frac{1}{d-1} \sum_{i=1}^{d-1} F_i$$

The first inequality follows since $h$ may also mislabel $x_0$. The second inquality follows because $N \sum_{i=1}^{d-1}[r(x_i) = 1] \leq Z$. The third inequality follows because $\sum_{i=1}^{d-1} \mathbf{1}[r(x_i) = 1]F_i$ is less than both $\sum_{i=1}^{d-1} F_i$ and $\sum_{i=1}^{d-1} \mathbf{1}[r(x_i) = 1]$.

By definition $\beta(1/2 - \lambda) = \eta$ and $\lambda\beta = 2\epsilon$. Therefore:

$$\mathbb{E}[L(h(x), y) \mid r(x) = 0] \geq \frac{1}{(1 - Z)}\left[\frac{\rho}{2} - \frac{Z}{2} + (1 - \rho)\eta + (1 - \rho)\frac{2\epsilon}{d-1}\sum_{i=1}^{d-1} F_i\right]$$

$$\geq \eta + \frac{1 - \rho}{1 - Z}\frac{2\epsilon}{d-1}\sum_{i=1}^{d-1} F_i \geq \eta + \frac{\epsilon}{d-1}\sum_{i=1}^{d-1} F_i \tag{1}$$

The second inequality above can be verified by some algebra when $\eta \leq 1/2$. In particular, $\frac{\frac{\rho-Z}{2} + \eta - \eta\rho}{1 - Z} \geq \eta \Leftrightarrow \frac{\rho-Z}{2} + \eta \geq (\rho - Z)\eta + \eta \Leftrightarrow \eta \leq 1/2$ when $Z \leq \rho$. The third inequality follows since $\rho \leq 1/2$ and so $\frac{1-\rho}{1-Z} \geq (1 - \rho) \geq 1/2$.

Since our distribution places $\rho$ mass on $\hat{x}$, it's clear that $r_0$ is the only function in $\mathcal{R}_\rho$ that is not in $\mathcal{R}'_\rho$.

Under the requester $r_0$, which avoids loss on $\hat{x}$, a hypothesis that correctly labels $x_0$ incurs a loss of:

$$\mathbb{E}[L(h(x), y \mid r_0(x) = 0] = \frac{1}{1 - \rho}\left[N \sum_{i=1}^{d-1} G_i(1/2 - \lambda) + N \sum_{i=1}^{d-1} F_i(1/2 + \lambda)\right]$$

$$= \frac{1}{1 - \rho}\left[(d - 1)N(\frac{1}{2} - \lambda) + N \sum_{i=1}^{d-1} 2\lambda F_i\right]$$

$$= \beta(1/2 - \lambda) + \frac{2\lambda\beta}{d-1}\sum_{i=1}^{d-1} F_i = \eta + \frac{4\epsilon}{d-1}\sum_{i=1}^{d-1} F_i. \tag{2}$$

Any hypothesis that incorrectly labels $x_0$ is strictly worse, and so equations (1) and (2), confirm that the optimal hypothesis requester pair is indeed $(h^*, r_0)$, with a loss of $\eta$ (since equation (2 holds with equality). Moreover, any suboptimal hypothesis requester pair has a loss of more than $\eta + \frac{\epsilon}{d-1}\sum_{i=1}^{d-1} F_i$, the minimum of (1) and (2).

We can now leverage Theorem 12 of Beygelzimer et al. [2009] . The theorem states that any algorithm that queries $x_i, i \geq 1$ fewer than $c'd\eta^2/\epsilon^2$ times , for some constant $c'$, will incorrectly predict more than $1/4$ of the bits $b_i$ with probability at least $1/2$, and thus will output $(\hat{h}, \hat{r})$ with error at least

$\eta + \epsilon/4$ with probability at least $1/2$. Setting $\epsilon = 4\sqrt{\frac{d\eta}{T}}$ tells us that any algorithm satisfying condition (A) in the statement of the theorem, must query at at least $Q \geq \frac{c'}{16}\eta T = \Omega(\eta T)$ examples before time $T$.

We next prove that if $\rho \geq 2c\eta$, then any algorithm satisfying condition (B) in the statement of the theorem must query $\Omega(\rho T)$ examples.

Recall that all $r \in \mathcal{R}'$, satisfy $r(\hat{x}) = 0$. Let $h$ be an arbitrary hypothesis satisfying $h(x_0) = 1$. The optimal $r' \in \mathcal{R}'$ for $h$, satisfies $r'(\hat{x}) = 0$, $r'(x_0) = 0$, $r'(x_i) = 1$ for all $i \geq 1$. In other words, $r'$ is forced to suffer loss on $\hat{x}$ because $r' \in \mathcal{R}'$, but avoids all other error otherwise.

$$
\begin{aligned}
\mathbb{E}[L(h(x), y) \mid r'(x) = 0] &= \mathbb{E}[L(h(x), y) \mid r'(x) = 0, x = \hat{x}]\mathbb{P}[x = \hat{x} \mid r'(x) = 0] \\
&\quad + \mathbb{E}[L(h(x), y) \mid r'(x) = 0, x = x_0]\mathbb{P}[x = x_0 \mid r'(x) = 0] \\
&= \mathbb{E}[L(h(x), y) \mid r'(x) = 0, x = \hat{x}]\mathbb{P}[x = \hat{x} \mid r'(x) = 0] \\
&= \frac{1}{2}\frac{\mathbb{P}(x = \hat{x}, r'(x) = 0)}{\mathbb{P}(r'(x) = 0)} = \frac{1}{2}\frac{\rho}{\rho + (1 - \rho)(1 - \beta)} \\
&\geq \frac{c\eta}{1 - \beta + \rho\beta} > c\eta
\end{aligned}
$$

Since $\mathbb{E}[L(h(x), y) \mid r'(x) = 0] > c\eta$, there is a sufficiently large $T$ such that $c\eta + \sqrt{\frac{2d\eta}{T}} < \mathbb{E}[L(h(x), y) \mid r'(x) = 0]$. Thus any $t > T/2$ is a failed round if the algorithm plays a requesting strategy in $\mathcal{R}'$. Any algorithm satisfying (B) in the statement of the theorem must therefore with probability at least $1/2$ select strategies $R_t \notin \mathcal{R}'$, satisfying $R_t(\hat{x}) = 1$ for all rounds $t > T/2$. For each such $R_t$, $\mathbb{E}[R_t(x) = 1] \geq \rho$, and thus $Q = \Omega(\rho T)$ when $\rho \geq 2c\eta$, completing the proof. $\square$

## B    Proofs of Generalization and Label Complexity Guarantees

In this appendix, we first prove generalization guarantees and then label complexity guarantees. For simplicity below, let $t_0$ be the first time $t$ such that $t \geq 16 \log(t/\delta)$, let $\Delta_t = \sqrt{\frac{\log(3t/\delta)}{t}}$, $\Delta_t' = (\frac{4}{1-\rho} + 2)\sqrt{\frac{2\log(6(t-t_0)(t-t_0+1)|\mathcal{H}\times\mathcal{R}|^2/\delta)}{t-t_0}}$ for any $t > t_0$, and $\tilde{\Delta}_t = 2\Delta_t' + \frac{8}{1-\rho}\Delta_{t-1}$.

### B.1    Generalization Guarantees

**Lemma 1.** *Assume $\rho < \frac{1}{2}$. For any $\delta > 0$, with probability at least $1 - \delta$, for all $s \geq 16 \log(3s/\delta)$, and any $r \in \mathcal{R}$,*

$$
\frac{1}{\mathbb{E}[r(x) = 0] + \Delta_{s-1}} \leq \frac{1}{\frac{1}{s-1}\sum_{s'=1}^{s-1} \mathbf{1}[r(x_{s'}) = 0]} \leq \frac{1}{\mathbb{E}[r(x) = 0] - \Delta_{s-1}}.
$$

*Proof.* By Hoeffding's inequality, $\mathbb{E}[r'(x) = 0] \leq \hat{\mathbb{E}}_{s-1}[\mathbf{1}[r'(x) = 0]] + \Delta_{s-1}$ and $\mathbb{E}[r'(x) = 0] \geq \hat{\mathbb{E}}_{s-1}[\mathbf{1}[r'(x) = 0]] - \Delta_{s-1}$ hold concurrently with probability at least $1 - \delta$.

The inequality $\mathbb{E}[r'(x) = 1] \leq \rho$ directly implies that $\mathbb{E}[r'(x) = 0] > 1 - \rho \geq \frac{1}{2}$. It also holds by assumption that $s - 1 \geq 16 \log(3(s-1)/\delta)$ which can be rewritten as $\frac{1}{2} - \Delta_{s-1} > \frac{1}{4}$. Hence, $\mathbb{E}[r'(x) = 0] - \Delta_{s-1} > \frac{1}{2} - \Delta_{s-1} > \frac{1}{4} > 0$. The statement of the lemma follows by inverting the concentration inequalities and then dividing by $\mathbb{E}[r'(x) = 0] - \Delta_{s-1}$ and by $\mathbb{E}[r'(x) = 0] + \Delta_{s-1}$, separately. $\square$

**Lemma 2.** *Assume $\rho < \frac{1}{2}$. For any $\delta > 0$, with probability at least $1 - \delta$, for all $s \geq 16 \log(3s/\delta)$, and any $(h, r), (h', r') \in \mathcal{H} \times \mathcal{R}$,*

$$
\left| \frac{L(h(x_s), y_s)\mathbf{1}[r(x_s) = 0]}{\frac{1}{s-1}\sum_{s'=1}^{s-1} \mathbf{1}[r(x_{s'}) = 0]} - \frac{L(h'(x_s), y_s)\mathbf{1}[r'(x_s) = 0]}{\frac{1}{s-1}\sum_{s'=1}^{s-1} \mathbf{1}[r'(x_{s'}) = 0]} \right| \leq \frac{4}{1 - \rho} + 2.
$$

*Proof.* By an application of Lemma 1, with probability at least $1 - \delta$,

$$\frac{L(h(x_s), y_s)\mathbf{1}[r(x_s) = 0]}{\frac{1}{s-1}\sum_{s'=1}^{s-1}\mathbf{1}[r(x_{s'}) = 0]} - \frac{L(h'(x_s), y_s)\mathbf{1}[r'(x_s) = 0]}{\frac{1}{s-1}\sum_{s'=1}^{s-1}\mathbf{1}[r'(x_{s'}) = 0]}$$

$$\leq \frac{L(h(x_s), y_s)\mathbf{1}[r(x_s) = 0]}{\mathbb{E}[r(x) = 0] - \Delta_{s-1}} - \frac{L(h'(x_s), y_s)\mathbf{1}[r'(x_s) = 0]}{\mathbb{E}[r'(x) = 0] + \Delta_{s-1}}$$

$$\leq \frac{\mathbb{E}[r(x) = 0] - \mathbb{E}[r'(x) = 0] + 2\Delta_{s-1}}{(\mathbb{E}[r'(x) = 0] + \Delta_{s-1})(\mathbb{E}[r(x) = 0] - \Delta_{s-1})}$$

$$= \frac{\mathbb{E}[r(x) = 0] - \mathbb{E}[r'(x) = 0]}{(\mathbb{E}[r'(x) = 0] + \Delta_{s-1})(\mathbb{E}[r(x) = 0] - \Delta_{s-1})} \tag{3}$$

$$+ \frac{2\Delta_{s-1}}{(\mathbb{E}[r'(x) = 0] + \Delta_{s-1})(\mathbb{E}[r(x) = 0] - \Delta_{s-1})} \tag{4}$$

where we used the fact that $L(h'(x_s), y_s)\mathbf{1}[r'(x_s) = 0] \leq 1$. First, we analyze the term (3). By the assumptions, it holds that $\mathbb{E}[r'(x) = 1] \leq \rho$ which implies $\mathbb{E}[r'(x) = 0] > 1 - \rho \geq \frac{1}{2}$ and that $s - 1 \geq 16\log(3(s-1)/\delta)$ which can be rewritten as $\frac{1}{2} - \Delta_{s-1} > \frac{1}{4}$. Hence, $\mathbb{E}[r'(x) = 0] - \Delta_{s-1} > \frac{1}{2} - \Delta_{s-1} > \frac{1}{4} > 0$. Thus,

$$(3) \leq \frac{4}{1 - \rho},$$

where we also used the fact that $\mathbb{E}[r'(x) = 0] + \Delta_{s-1} \geq 1 - \rho$.

Next, we turn to term (4). For simplicity, let $a = \mathbb{E}[r(x) = 0]$ and $b = \mathbb{E}[r'(x) = 0]$ and consider the following

$$(a + \Delta_{s-1})(b - \Delta_{s-1})/\Delta_{s-1} = ab/\Delta_{s-1} - a + b - \Delta_{s-1}$$

$$> \frac{1}{4\Delta_{s-1}} - 1 + \frac{1}{4} = \frac{1}{4\Delta_{s-1}} - \frac{3}{4} > 1$$

where we used the fact that $b - \Delta_{s-1} > 1/4$, $ab > \frac{1}{4}$, and $a < 1$ by the same reasoning as term (3) and that $1/\Delta_{s-1} - 3 > 1$ since by assumption $\frac{1}{4} > \Delta_{s-1}$. Hence,

$$(4) = \frac{2}{ab/\Delta_{s-1} - a + b - \Delta_{s-1}} \leq 2.$$

Putting the above together, $(3) + (4) \leq \frac{4}{1-\rho} + 2$ and taking absolute values concludes the proof. $\square$

**Theorem 2.** *Given any $\rho < \frac{1}{2}$, for any $\delta > 0$, with probability at least $1 - \delta$, for all $t \geq 16\log(3t/\delta)$,*
$$\mathbb{E}[L(h_t(x), y)|r_t(x) = 0] \leq \mathbb{E}[L(h^*(x), y)|r^*(x) = 0] + \tilde{O}\big(\sqrt{(1/t)\log(1/\delta)}\big).$$

*Proof.* Recall that $t_0$ is the first time $t$ such that $t \geq 16\log(t/\delta)$ and consider only $t > t_0$. For any pair $(h, r) \in V_t$ and $(h', r') \in V_t$, let

$$Z_s = \mathbb{E}\left[\frac{q_s}{p_s(x_s)}\frac{L(h(x_s), y_s)\mathbf{1}[r(x_s) = 0]}{\frac{1}{s-1}\sum_{s'=1}^{s-1}\mathbf{1}[r(x_{s'}) = 0]}\right] - \mathbb{E}\left[\frac{q_s}{p_s(x_s)}\frac{L(h'(x_s), y_s)\mathbf{1}[r'(x_s) = 0]}{\frac{1}{s-1}\sum_{s'=1}^{s-1}\mathbf{1}[r'(x_{s'}) = 0]}\right]$$

$$- \frac{q_s}{p_s(x_s)}\left(\frac{L(h(x_s), y_s)\mathbf{1}[r(x_s) = 0]}{\frac{1}{s-1}\sum_{s'=1}^{s-1}\mathbf{1}[r(x_{s'}) = 0]} - \frac{L(h'(x_s), y_s)\mathbf{1}[r'(x_s) = 0]}{\frac{1}{s-1}\sum_{s'=1}^{s-1}\mathbf{1}[r'(x_{s'}) = 0]}\right),$$

for $s \in [t]$. If $p_s < 1$, then $Z_s \leq 2$ by definition. If $p_s = 1$, then $|Z_s| \leq \frac{8}{1-\rho} + 4$ follows by Lemma 2 with probability at least $1 - \delta$. Under this high probability event, we apply Azuma's inequality to $Z_s$ to attain:

$$\left|\frac{1}{t - t_0}\sum_{s=t_0}^{t}\mathbb{E}\left[\frac{q_s}{p_s(x_s)}\frac{L(h(x_s), y_s)\mathbf{1}[r(x_s) = 0]}{\frac{1}{s-1}\sum_{s'=1}^{s-1}\mathbf{1}[r(x_{s'}) = 0]}\right] - \frac{1}{t - t_0}\sum_{s=t_0}^{t}\mathbb{E}\left[\frac{q_s}{p_s(x_s)}\frac{L(h'(x_s), y_s)\mathbf{1}[r'(x_s) = 0]}{\frac{1}{s-1}\sum_{s'=1}^{s-1}\mathbf{1}[r'(x_{s'}) = 0]}\right]\right.$$

$$\left.- \frac{1}{t - t_0}\sum_{s=t_0}^{t}\frac{q_s}{p_s(x_s)}\left(\frac{L(h(x_s), y_s)\mathbf{1}[r(x) = 0]}{\frac{1}{s-1}\sum_{s'=1}^{s-1}\mathbf{1}[r(x_{s'}) = 0]} - \frac{L(h'(x_s), y_s)\mathbf{1}[r'(x) = 0]}{\frac{1}{s-1}\sum_{s'=1}^{s-1}\mathbf{1}[r'(x_{s'}) = 0]}\right)\right| \leq 2\Delta_t',$$

where we take a union bound over all $(h, r) \in V_t$. Equivalently, this can be written as:

$$\left| \mathbb{E}[\widehat{L}_t(h, r)] - \mathbb{E}[\widehat{L}_t(h', r')] - \widehat{L}_t(h, r) - \widehat{L}_t(h', r') \right| \le 2\Delta'_t, \tag{5}$$

To attain our desired bound, we then relate $\mathbb{E}[\widehat{L}_t(h, r)]$ to $\mathbb{E}[L(h(x), y)|r(x) = 0]$. First, we rewrite the former expectation:

$$\mathbb{E}[\widehat{L}_t(h, r)] = \frac{1}{t - t_0} \sum_{s=t_0}^{t} \mathbb{E}\left[ \frac{q_s}{p_s(x_s)} \frac{L(h(x_s), y_s)\mathbf{1}[r(x_s) = 0]}{\frac{1}{s-1} \sum_{s'=1}^{s-1} \mathbf{1}[r(x_{s'}) = 0]} \right]$$

$$= \frac{1}{t - t_0} \sum_{s=t_0}^{t} \mathbb{E}\left[ \frac{L(h(x_s), y_s)\mathbf{1}[r(x_s) = 0]}{\frac{1}{s-1} \sum_{s'=1}^{s-1} \mathbf{1}[r(x_{s'}) = 0]} \right]$$

$$= \mathbb{E}[L(h(x), y)\mathbf{1}[r(x) = 0]] \frac{1}{t - t_0} \sum_{s=t_0}^{t} \mathbb{E}\left[ \frac{1}{\frac{1}{s-1} \sum_{s'=1}^{s-1} \mathbf{1}[r(x_{s'}) = 0]} \right] \tag{6}$$

where the first inequality follows since $\mathbb{E}[q_s|p_s(x_s)] = p_s(x_s)$ and where the second inequality follows by the fact that the data is i.i.d. Combining this with Lemma 1 multiplied by $\mathbb{E}[L(h(x), y)\mathbf{1}[r(x) = 0]]$, it follows that

$$\mathbb{E}[L(h(x), y)\mathbf{1}[r(x) = 0]] \frac{1}{t - t_0} \sum_{s=t_0}^{t} \mathbb{E}\left[ \frac{1}{\mathbb{E}[r(x) = 0] + \Delta_{s-1}} \right] \tag{7}$$

$$\le \mathbb{E}[\widehat{L}_t(h, r)]$$

$$\le \mathbb{E}[L(h(x), y)\mathbf{1}[r(x) = 0]] \frac{1}{t - t_0} \sum_{s=t_0}^{t} \mathbb{E}\left[ \frac{1}{\mathbb{E}[r(x) = 0] - \Delta_{s-1}} \right]. \tag{8}$$

Hence, in order to relate $\mathbb{E}[\widehat{L}_t(h, r)]$ to $\mathbb{E}[L(h(x), y)|r(x) = 0]$, we also need that $\frac{1}{\mathbb{E}[r(x)=0]\pm\Delta_{s-1}}$ is close to $\frac{1}{\mathbb{E}[r(x)=0]}$ and so consider the following:

$$\left| \frac{1}{t - t_0} \sum_{s=t_0}^{t} \left( \frac{1}{\mathbb{E}[r(x) = 0] - \Delta_{s-1}} - \frac{1}{\mathbb{E}[r(x) = 0]} \right) \right| = \frac{1}{t - t_0} \sum_{s=t_0}^{t} \frac{\Delta_{s-1}}{(\mathbb{E}[r(x) = 0] - \Delta_{s-1})\, \mathbb{E}[r(x) = 0]}$$

$$= \frac{1}{\mathbb{E}[r(x) = 0]} \frac{1}{t - t_0} \sum_{s=t_0}^{t} \frac{\Delta_{s-1}}{(\mathbb{E}[r(x) = 0] - \Delta_{s-1})]}$$

$$\le \frac{1}{\mathbb{E}[r(x) = 0]} \left( 4\frac{\Delta_{t-1}}{\mathbb{E}[r(x) = 0]} \right) \tag{9}$$

where in the last inequality we used the fact that $\mathbb{E}[r'(x) = 0] - \Delta_{s-1} > \frac{1}{2} - \Delta_{s-1} > \frac{1}{4}$ via the same reasoning as in Lemma 1. Similarly,

$$\left| \frac{1}{t - t_0} \sum_{s=t_0}^{t} \left( \frac{1}{\mathbb{E}[r(x) = 0] + \Delta_{s-1}} - \frac{1}{\mathbb{E}[r(x) = 0]} \right) \right| = \frac{1}{t - t_0} \sum_{s=t_0}^{t} \frac{\Delta_{s-1}}{(\mathbb{E}[r(x) = 0] + \Delta_{s-1})\, \mathbb{E}[r(x) = 0]}$$

$$= \frac{1}{\mathbb{E}[r(x) = 0]} \frac{1}{t - t_0} \sum_{s=t_0}^{t} \frac{\Delta_{s-1}}{(\mathbb{E}[r(x) = 0] + \Delta_{s-1})}$$

$$\le \frac{1}{\mathbb{E}[r(x) = 0]} \left( 4\frac{\Delta_{t-1}}{\mathbb{E}[r(x) = 0]} \right). \tag{10}$$

Multiplying (9) and (10) by $\mathbb{E}[L(h(x), y)\mathbf{1}[r(x) = 0]]$ and using the fact that $\mathbb{E}[r(x) = 0] \ge 1 - \rho$ and $\mathbb{E}[L(h(x), y)|r(x) = 0] \le 1$, it follows that

$$\left| \mathbb{E}[L(h(x), y)\mathbf{1}[r(x) = 0]] \frac{1}{t - t_0} \sum_{s=t_0}^{t} \mathbb{E}\left[ \frac{1}{\mathbb{E}[r(x) = 0] + \Delta_{s-1}} \right] - \mathbb{E}[L(h(x), y)|r(x) = 0] \right|$$

$$\le \mathbb{E}[L(h(x), y)|r(x) = 0] \left( 4\frac{\Delta_{t-1}}{\mathbb{E}[r(x) = 0]} \right) \le \frac{4}{1 - \rho}\Delta_{t-1} \tag{11}$$

and

$$\left| \mathbb{E}[L(h(x),y)\mathbf{1}[r(x)=0]]\frac{1}{t-t_0}\sum_{s=t_0}^{t}\mathbb{E}\left[\frac{1}{\mathbb{E}[r(x)=0]-\Delta_{s-1}}\right] - \mathbb{E}[L(h(x),y)|r(x)=0]\right|$$

$$\leq \frac{4}{1-\rho}\Delta_{t-1}. \tag{12}$$

Using the series of inequalities (8), (11), and (12) into the result of the concentration inequality of (5) along with some algebra,

$$|\mathbb{E}[L(h(x),y)|r(x)=0] - \mathbb{E}[L(h'(x),y)|r'(x)=0] - \widehat{L}_t(h,r) + \widehat{L}_t(h',r')|$$

$$\leq 2\Delta'_t + \frac{8}{1-\rho}\Delta_{t-1}. \tag{13}$$

Recalling that $\tilde{\Delta}_t = 2\Delta'_t + \frac{8}{1-\rho}\Delta_{t-1}$. Since $V_t \subseteq V_{t-1}$, it holds that for any pair $(h,r) \in V_t$ and $(h',r') \in V_t$ via inequality (13):

$$\mathbb{E}[L(h(x),y)|r(x)=0] - \mathbb{E}[L(h'(x),y)|r'(x)=0]$$
$$\leq \widehat{L}_{t-1}(h,r) - \widehat{L}_{t-1}(h',r') + \tilde{\Delta}_{t-1}$$
$$\leq \min_{(h,r)\in V_{t-1}}\widehat{L}_{t-1}(h,r) + \tilde{\Delta}_{t-1} - \min_{(h,r)\in V_{t-1}}\widehat{L}_{t-1}(h,r) + \tilde{\Delta}_{t-1}$$
$$= 2\tilde{\Delta}_{t-1}.$$

We then prove that $(h^*,r^*) \in V_t$. By induction, suppose that $(h^*,r^*) \in V_{t-1}$. Let $(h'_{t-1},r'_{t-1}) = \min_{(h,r)\in V_{t-1}}\widehat{L}_{t-1}(h,r)$. Then, by inequality (13):

$$\widehat{L}_{t-1}(h^*,r^*)$$
$$\leq \widehat{L}_{t-1}(h'_{t-1},r'_{t-1}) - \mathbb{E}[L(h'_{t-1}(x),y)|r'_{t-1}(x)=0] + \mathbb{E}[L(h^*(x),y)|r^*(x)=0] + \tilde{\Delta}_{t-1}$$
$$\leq \widehat{L}_{t-1}(h'_{t-1},r'_{t-1}) + \tilde{\Delta}_{t-1}$$

which by definition means that $(h^*,r^*) \in V_t$. $\qquad\square$

**Corollary 1.** *Given any $\rho < \frac{1}{2}$, for any $\delta > 0$, with probability at least $1-\delta$, for all $t \geq 16\log(3t/\delta)$, $\mathbb{E}[L(h_t(x),y)|r_t(x)=0] \leq \mathbb{E}[L(h_b(x),y)] + \gamma + \tilde{O}\big(\sqrt{(1/t)\log(1/\delta)}\big)$, where $\gamma = \mathbb{E}[L(h^*(x),y)|r^*(x)=0] - \mathbb{E}[L(h_b(x),y)]$. Furthermore, if $r_\diamond \in \mathcal{R}_\rho$, then $\gamma \leq 0$.*

*Proof.* Since $h_b \in \mathcal{H}$ and since $r_\diamond \in \mathcal{R}$, it holds that $\mathbb{E}[L(h^*(x),y)|r^*(x)=0] \leq \mathbb{E}[L(h_b(x),y)]$. $\qquad\square$

## B.2   Label Complexity

**Proposition 1.** *For all $\delta > 0$, with probability at least $1-\delta$, for all $t \geq 16\log(3t/\delta)$,*

$$\mathbb{E}[p_s(x_s)] \leq 4\theta\,\mathbb{E}[L(h^*(x),y)|r^*(x)=0] + \tilde{O}\left(\theta\sqrt{\frac{\log(1/\delta)}{t}}\right),$$

*where $\theta$ is the disagreement coefficient.*

*Proof.* By Lemma (1) in conjunction with Inequalities (9) and (10), it holds that

$$\mathbb{E}[p_s(x_s)] \le \mathbb{E}\left[\max_{(h,r),(h',r')\in V_s} \max_y \frac{L(h(x_s),y)\mathbf{1}[r(x_s)=0]}{\frac{1}{s-1}\sum_{s'=1}^{s-1}\mathbf{1}[r(x_{s'})=0]} - \frac{L(h'(x_s),y)\mathbf{1}[r'(x_s)=0]}{\frac{1}{s-1}\sum_{s'=1}^{s-1}\mathbf{1}[r'(x_{s'})=0]}\right]$$

$$\le 2\,\mathbb{E}\left[\max_{(h,r)\in V_s}\max_y \frac{L(h(x_s),y)\mathbf{1}[r(x_s)=0]}{\mathbb{E}[r(x)=0]-\Delta_{s-1}} - \frac{L(h'(x_s),y)\mathbf{1}[r'(x_s)=0]}{\mathbb{E}[r'(x)=0]+\Delta_{s-1}}\right]$$

$$= 2\,\mathbb{E}\left[\max_{(h,r)\in V_s}\max_y \frac{L(h(x_s),y)\mathbf{1}[r(x_s)=0]}{\mathbb{E}[r(x)=0]-\Delta_{s-1}} - \frac{L(h(x_s),y)\mathbf{1}[r(x_s)=0]}{\mathbb{E}[r(x)=0]}\right.$$

$$+ \frac{L(h(x_s),y)\mathbf{1}[r(x_s)=0]}{\mathbb{E}[r(x)=0]} - \frac{L(h'(x_s),y)\mathbf{1}[r'(x_s)=0]}{\mathbb{E}[r'(x)=0]+\Delta_{s-1}} - \frac{L(h'(x_s),y)\mathbf{1}[r'(x_s)=0]}{\mathbb{E}[r'(x)=0]}$$

$$\left.+ \frac{L(h'(x_s),y)\mathbf{1}[r'(x_s)=0]}{\mathbb{E}[r'(x)=0]}\right]$$

$$\le \mathbb{E}\left[\max_{(h,r),(h',r')\in V_s}\max_y \left|\frac{L(h(x_s),y)\mathbf{1}[r(x_s)=0]}{\mathbb{E}[r(x)=0]} + \frac{L(h'(x_s),y)\mathbf{1}[r'(x_s)=0]}{\mathbb{E}[r'(x)=0]}\right|\right]$$

$$+ \frac{8}{(1-\rho)^2}\Delta_{s-1}.$$

We then focusing on bounding the first term above. By Theorem 2, $V_s \subseteq \{(h,r) \in \mathcal{H} \times \mathcal{R} : \mathbb{E}[L(h(x),y)|r(x)=0] \le \mathbb{E}[L(h^*(x),y)|r^*(x)=0]+2\tilde{\Delta}_{s-1}\}$. Using this fact in conjunction with $\rho((h,r),(h^*,r^*)) \le \mathbb{E}[L(h(x),y)|r(x)=0]+\mathbb{E}[L(h^*(x),y)|r^*(x)=0]$ implies that $V_s \subseteq B(h^*,r^*,\Lambda)$ where $\Lambda = 2\,\mathbb{E}[L(h^*(x),y)|r^*(x)=0]+2\tilde{\Delta}_{s-1}$. where we used the definition of disagreement coefficient in the last inequality.

Using the above it holds that:

$$\mathbb{E}\left[\max_{(h,r),(h',r')\in V_s}\max_y \left|\frac{L(h(x_s),y)\mathbf{1}[r(x_s)=0]}{\mathbb{E}[r(x)=0]} - \frac{L(h'(x_s),y)\mathbf{1}[r'(x_s)=0]}{\mathbb{E}[r'(x)=0]}\right|\right]$$

$$\le 2\,\mathbb{E}\left[\max_{(h,r)\in V_s}\max_y \left|\frac{L(h(x_s),y)\mathbf{1}[r(x_s)=0]}{\mathbb{E}[r(x)=0]} - \frac{L(h^*(x_s),y)\mathbf{1}[r^*(x_s)=0]}{\mathbb{E}[r^*(x)=0]}\right|\right]$$

$$\le 2\,\mathbb{E}\left[\max_{(h,r)\in B(h^*,r^*,\Lambda)}\max_y \left|\frac{L(h(x_s),y)\mathbf{1}[r(x_s)=0]}{\mathbb{E}[r(x)=0]} - \frac{L(h^*(x_s),y)\mathbf{1}[r^*(x_s)=0]}{\mathbb{E}[r^*(x)=0]}\right|\right]$$

$$\le 2\theta(2\,\mathbb{E}[L(h^*(x),y)|r^*(x)=0]+2\tilde{\Delta}_{s-1})$$

where we used the definition of disagreement coefficient in the last inequality. $\square$

**Theorem 3.** *Given any $\rho < \frac{1}{2}$, for all $\delta > 0$, with probability at least $1-\delta$, $\sum_{s=1}^T \mathbb{E}[p_s(x_s)] = \tilde{O}(\theta\eta T + \theta\sqrt{T})$, where $\theta$ is the disagreement coefficient.*

*Proof.* The theorem follows directly by summing Proposition 1 over the rounds. $\square$

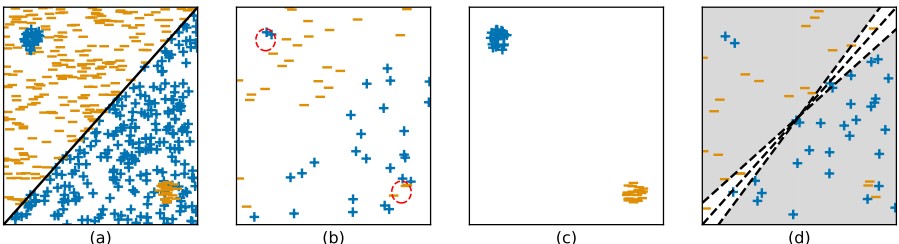

Figure 3: Example distribution: (a) A distribution in $\mathbb{R}^2$. (b) Region selected by an optimal abstention algorithm is depicted by the red circles. (c) Dataset induced by the abstention region in (b). (d) Disagreement region for active learning is shown in white.

## C  Comparisons to Active and Abstention Learning:

We now provide a simple example to help distinguish our setting from both active and abstention learning. Consider the distribution depicted in Figure 3 (a), and suppose the learner is given a hypothesis class consisting of halfspaces.

Figure 3 (b) depicts the optimal abstention region as dotted red circles. Suppose the learner were handed this region by an oracle (ignoring any computation and sample complexity concerns in actually finding this region). Then recall that in our setting the mechanism for avoiding loss (i.e., "abstaining") is also the same mechanism for generating labeled examples. Thus, if the learner were to solely request labels from the optimal abstention region, it would generate the dataset depicted in Figure 3 (c) and this dataset would induce precisely the wrong hypothesis.

Figure 3 (d) depicts a finite dataset drawn according to the distribution and the dotted lines correspond to a set of "good" hypotheses. Active learning algorithms generally label examples as long as there is disagreement between good hypotheses on the label of the example. While the exact details vary by algorithm, an optimal active learner gains no information by labeling an example that all good hypotheses agree on. Thus, the grayed region in Figure 3 (d) will never be labeled. Moreover, as the active learner hones in on the optimal hypothesis, and the set of candidate good hypotheses shrinks, it will eventually stop requesting labels entirely, even though there is value in requesting labels purely for evading the loss of an incorrect prediction (e.g. requesting labels inside the two dotted red circles).

| dataset | # features | # train | # test | $|H|$ | $1/C$ |
|---|---|---|---|---|---|
| a9a | 123 | 20,000 | 12,561 | 64 | $2^{-11}$ |
| cod-rna | 8 | 25,000 | 34,535 | 64 | $2^{-13}$ |
| mnist | 780 | 30,000 | 30,000 | 32 | $2^{-10}$ |
| phishing | 68 | 9,000 | 2,055 | 46 | $2^{-11}$ |
| shuttle | 9 | 20,000 | 23,499 | 64 | $2^{-13}$ |
| skin | 3 | 150,000 | 95,065 | 46 | $2^{-11}$ |

Table 1: Dataset characteristics.

## D  Empirical Investigation Appendix

In this section, we first give additional details on the DPL-IWAL ablation study, then present a study on estimating $\gamma$ for margin-based requesters, and finally present additional details for the evaluation of DPL-Simplified.

### D.1  DPL-IWAL Ablation Study

In this study, we test the following publicly available datasets: `mnist`, `shuttle`, `cod-rna`, `phishing`, `skin` and `a9a`. Details of dataset training/test split sizes, number features, and the size of finite hypothesis sets used in the ablation study is shown in Table 1. All datasets can be found at the LIBSVM dataset repository: https://www.csie.ntu.edu.tw/~cjlin/libsvmtools/datasets/, released under the 3-clause BSD license. For the `mnist` dataset we learn the binary classification of odd vs. even digits and for the `shuttle` dataset we classify the majority class vs. the rest. For all datasets we normalize features by first centering each feature (subtracting the mean value of each feature column) and then scaling to unit variance (dividing by the standard deviation of each feature column). After this, the entire data matrix is scaled uniformly so that the maximum instance feature vector has unit norm, i.e. $\max_i \|x_i\| = 1$.

For the $\mathcal{H}$ and $\mathcal{R}_\rho$ set construction, we use logistic regression models trained using the `scikit-learn` library with `solver="liblinear"` and with L2 regularization parameter $C$ set as indicated in the table. The hypotheses are each trained using a small random sample of data, with a size uniformly selected between 30 and 500 data points. Then to create the margin-based requesters $\mathcal{R}_\rho$, for each hypothesis $h \in \mathcal{H}$, we use the *unlabeled* training fold examples to estimate a threshold $\tau$ that captures $\rho$ fraction of the distribution.

For each of the baseline methods, we adjust the uniform sampling rate so that the overall number of requested labels matches that of our proposed algorithm. See Figure 5 for the labeling budget as well as a breakdown of whether the request came from $q_t$ and/or $r_t$. Similar to IWAL Beygelzimer et al. [2009], the sampling probability $p_t$ of DPL-IWAL can be adjusted by rescaling the loss function by a constant. In these experiments, we found it effective to downweight the loss difference of Algorithm 1 by a factor 0.1 when the signs of $h(x)$ and $h'(x)$ agree. In order to increase efficiency, at some cost in adaptivity, we update the version space and best model/requester pair after every 25 examples, which is a standard technique when applying active learning in practice Amin et al. [2020].

Figure 6 shows the results of our ablation study on all datasets. We observe that the uniform random $r_t$ attains similar test error to the fully uniform baseline. This is because if $r$ is a uniform random requestor, then for all $h \in \mathcal{H}$, it holds that $\mathbb{E}[L(h(x), y)|r(x) = 0] = \mathbb{E}[L(h(x), y)]$. Thus, uniform $r_t$ behaves like the standard active learner, IWAL, which will generalize as well as a passive learner, meaning that with respect to the number of points processed, it will attain a test error close to that of a passive learner. At the same time, online errors for uniform $r_t$ are smaller than that of the fully uniform baseline. This suggests that the samples that are selected by an active learning strategy are also somewhat correlated with mistakes. Selecting a non-trivial requestor, but constraining $q_t$ to sample uniformly demonstrates that there is indeed significant value to selecting a good request region, as demonstrated by lower online mistakes and smaller conditional test errors.

Our experiments with the uniform $r_t$ baseline demonstrate that naively deploying IWAL in a dual purpose setting will yield suboptimal results. This shows the considerable benefits of DPL-IWAL, which uses an IWAL inspired sampling scheme carefully adapted for the dual purpose setting. A natural question is whether DPL-IWAL can use different active learning algorithms for its sampling

| dataset | # features | # examples | $1/C$ | notes |
|---------|-----------|-----------|-------|-------|
| a9a | 123 | 32,561 | $10^{-3}$ | |
| acoustic | 50 | 78,823 | $10^{-4}$ | class 1 vs. rest |
| cifar | 10 | 60,000 | $10^{-3}$ | project to 10-dim w/ PCA; class 1 vs. rest |
| cod-rna | 8 | 59,535 | $10^{-3}$ | |
| covtype | 54 | 581,012 | $10^{-5}$ | called "covtype.binary" on LIBSVM site |
| HIGGS | 28 | 100,000 | $10^{-4}$ | randomly subsampled from full 11M dataset |
| ijcnn | 22 | 49,990 | $10^{-4}$ | |
| mnist | 780 | 60,000 | $10^{-2}$ | odd vs. even |
| shuttle | 9 | 43,499 | $10^{-5}$ | class 1 vs. rest |
| skin | 3 | 245,065 | $10^{-2}$ | |

Table 2: Dataset characteristics.

procedure (such as uncertaity/margin sampling, DHM, or query-by-committee Balcan et al. [2007], Dasgupta et al. [2008], Dagan and Engelson [1995]), and whether these result in even more effective algorithms for the dual purpose setting. These directions pose new technical challenges, for example, if using a margin-based sampler $q_t$ in addition to a margin-based requestor $r_t$, the two of which may be highly correlated, it becomes non-trivial to estimate the loss conditioned on $r_t = 0$ using samples selected by $q_t$. Nevertheless, we look forward to investigating these potentially fruitful directions in future work.

Additionally, in Figure 7, we measure the validity of the assumption $E[L(h_t(x), y)|R_t(x) = 0] \leq E[L(h_t(x), y)|r_t(x) = 0]$, which is used in the discussion of the label complexity upper bound. This figures shows that cross these benchmark datasets, the assumption clearly holds.

### D.2 Estimating $\gamma$ for Margin-based Requestors

Recall that in Corollary 1 the bound on the mistakes of our algorithm's chosen hypothesis as compared to the best-in-class is more favorable the more negative $\gamma$ is. In this study, we measure an upper bound on the value of $\gamma$ for a margin-based requester function $r$ used in the ablation study and use a continuous hypothesis space to estimate $h_b = \operatorname{argmin}_h \mathbb{E}[L(h(x), y)]$.

We can measure a (potentially pessimistic) upper bound on $\gamma$ for the class $\mathcal{R}_{\rho, \mathcal{H}}$ by using: $\gamma = \mathbb{E}[L(h^*(x), y) \mid r^*(x) = 0] - \mathbb{E}[L(h_b(x), y)] \leq \mathbb{E}[L(h_b(x), y) \mid r_{h_b}(x) = 0] - \mathbb{E}[L(h_b(x), y)] = \mathbb{E}[L(h_b(x), y) \mid |h_b(x)| > \tau_{h, \rho}(\mathcal{D}_\mathcal{X})] - \mathbb{E}[L(h_b(x), y)]$. For each dataset, we first estimate $h_b$ minimizing the log-loss over the full dataset, then we measure the empirical estimate of $\mathbb{E}[L(h_b(x), y) \mid |h_b(x)| > \tau]$ for various choices of $\tau$. In Figure 4, we show measurements of this upper bound on $\gamma$ for varying values of the $C$ regularization parameter. The left-most data-point of the plot is the expected loss of $h_b$, that is $\mathbb{E}[L(h_b(x), y) \mid |h_b(x)| > 0] = \mathbb{E}[L(h_b(x), y)]$. As the graph progresses to the right, for each non-zero threshold value, the conditional loss outside of the threshold is always smaller than the value at $\tau = 0$. This implies a strictly negative upper bound on the value of $\gamma$. We also observe that $\gamma$ tends to grow as $\tau$, or equivalently $\rho$, increases. This empirical finding of a strictly negative $\gamma$ verifies the hypothesis returned by our algorithm admits a favorable bound in Corollary 1 as compared to the best-in-class.

### D.3 DPL-Simplified Study

In Section 6 we evaluate the DPL-Simplified algorithm and baselines using the 10 datasets described in Table 2, all of which can be found on the LIBSVM dataset website: https://www.csie.ntu.edu.tw/~cjlin/libsvmtools/datasets/. Some of the datasets which are multi-class case into binary classification problems, which is indicated in the notes column of Table 2 (along with any other pre-processing).

For all datasets we normalize features by first centering each feature (subtracting the mean value of each feature column) and then scaling to unit variance (dividing by the standard deviation of each feature column). After this, the entire data matrix is scaled uniformly so that the maximum instance feature vector has unit norm, i.e. $\max_i \|x_i\| = 1$.

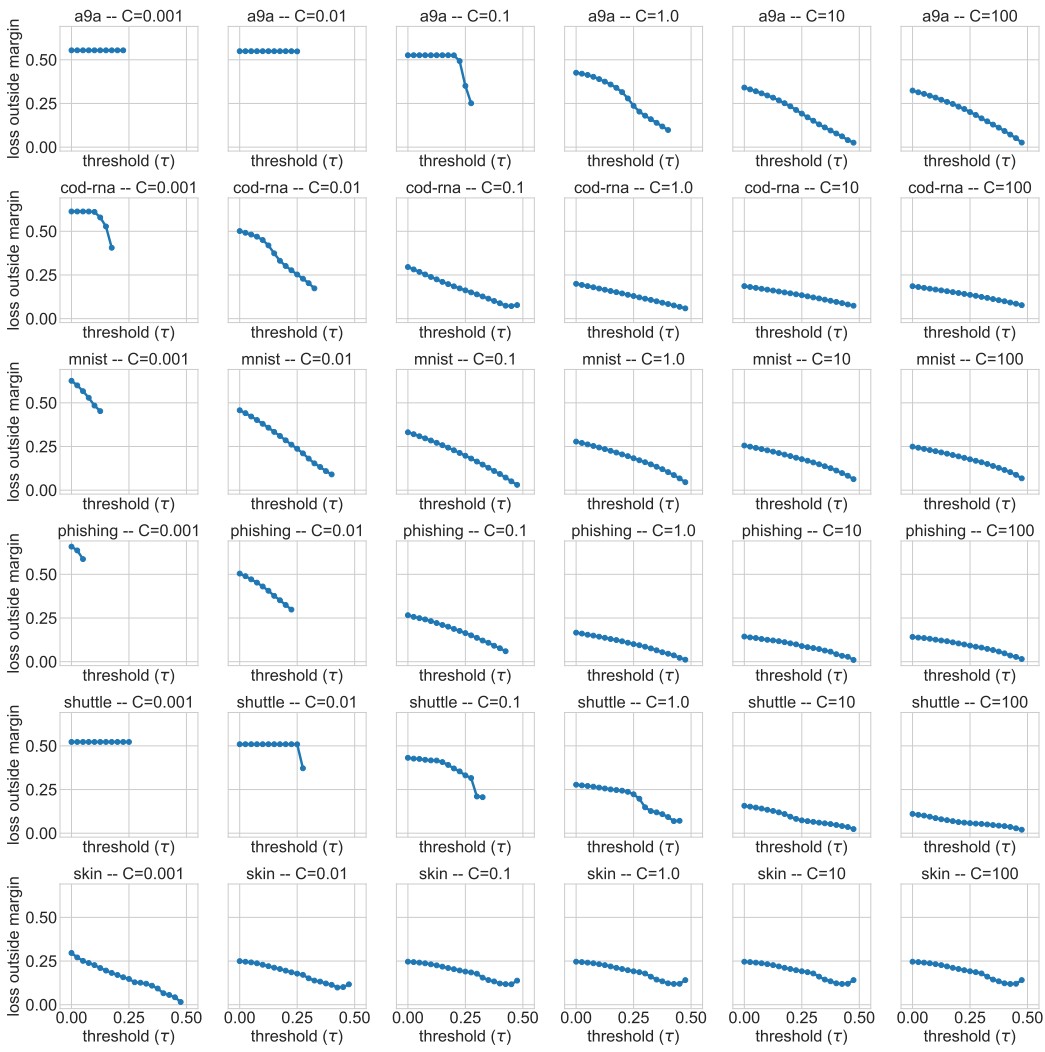

Figure 4: Empirical estimate of $\mathbb{E}[L(h_b(x), y) \mid |h_b(x)| > \tau]$ as a function of $\tau$, which can be used to lower bound $\gamma$. As explained in the text, the fact that the value at $\tau = 0$ is strictly larger than the values at any $\tau > 0$, implies a strictly negative upper bound on $\gamma$ for the linear model family, $\mathcal{H}$, and margin-based requester class $\mathcal{R}_{\rho, \mathcal{H}}$, across these distributions.

The scikit-learn library is used to train the logistic regression model, $h$, using `solver='saga'`, `max_iter=10000`, and setting the $L_2$ regularization parameter indicated in Table 2 for each dataset. In order to train the KNN model needed for the KNN- and Mixture-Requesters, we use scikit-learn's `KNeighborsClassifier`, with `n_neighbors=10`, `weights='distance'`, and `algorithm='brute'`.

Finally, Figure 8 shows the full learning curves associated with Table 2 in the main paper.

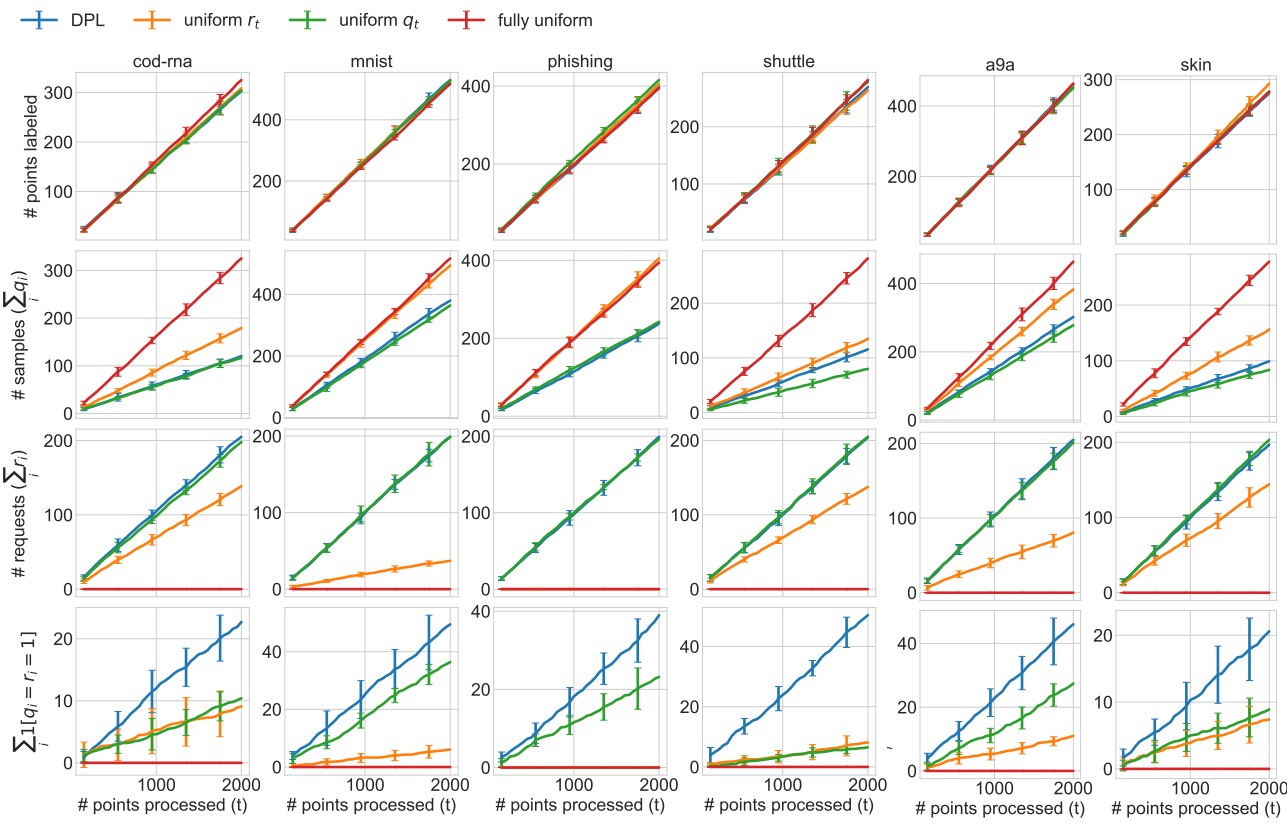

Figure 5: This figure shows for each dataset (each column) the overall number of labels requested throughout the training process (first row), the number examples that where $q_t = 1$ (second row), the number of examples where $r_t(x_t) = 1$ (third row), an finally, the number of cases where are label was requested due to both $q_t$ and $r_t(x_t)$ making a request.

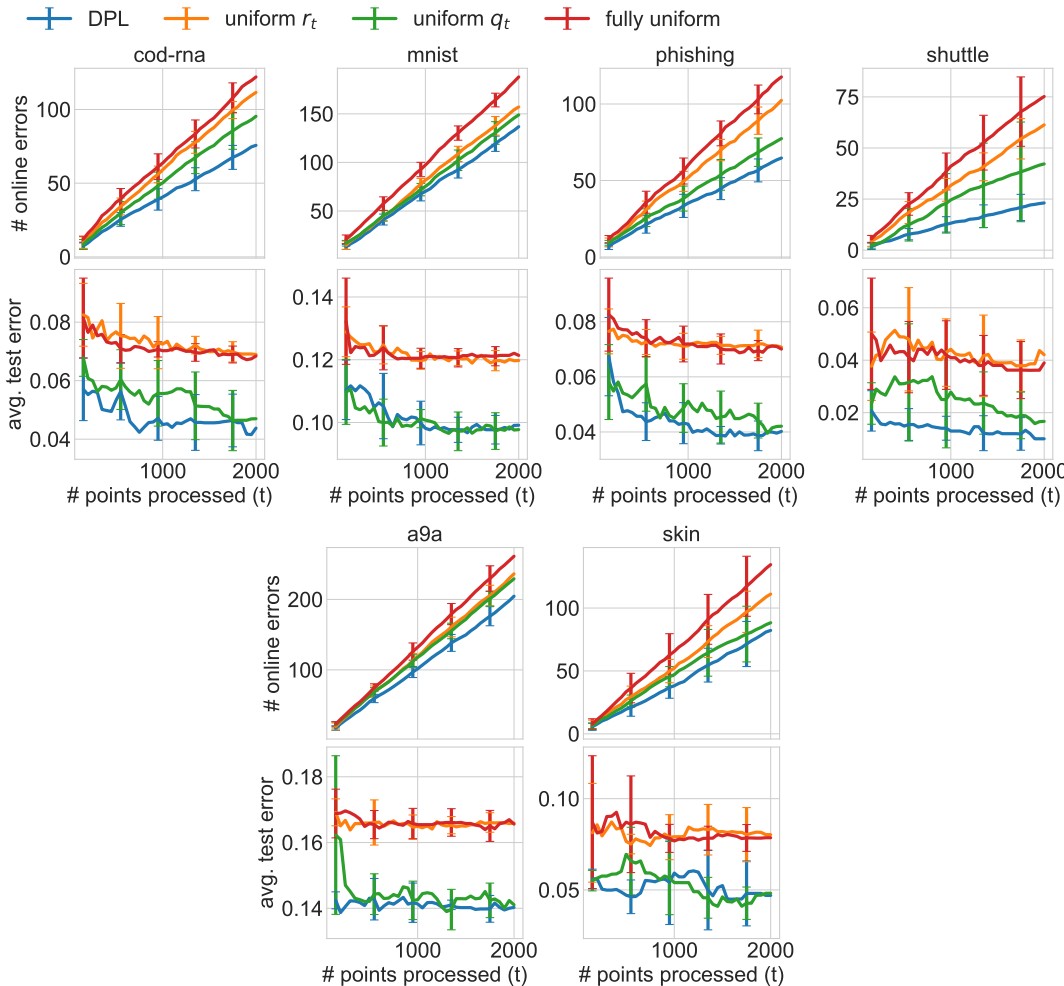

Figure 6: The number of online mistakes made while processing a stream of data as well as the held-out conditional loss on non-requested points made by DPL-IWAL and baselines comparators. The plots show the mean and standard deviation over 10 trials.

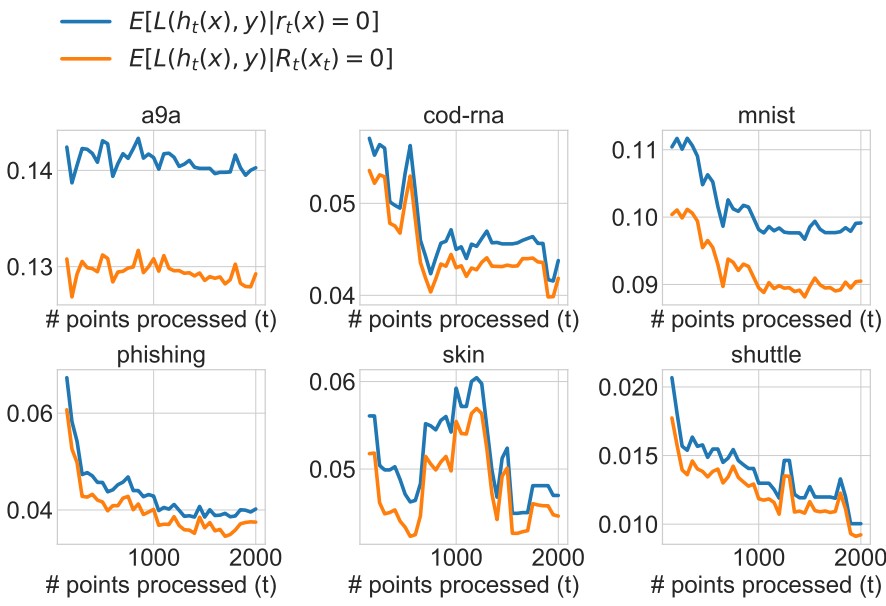

Figure 7: The blue curve is equivalent to the conditional test error of the DPL-IWAL algorithm, while the orange curve display the condition loss on the set of points where $R_t(x) = 0$, i.e. $q_t = 0 \wedge r_t(x) = 0$.

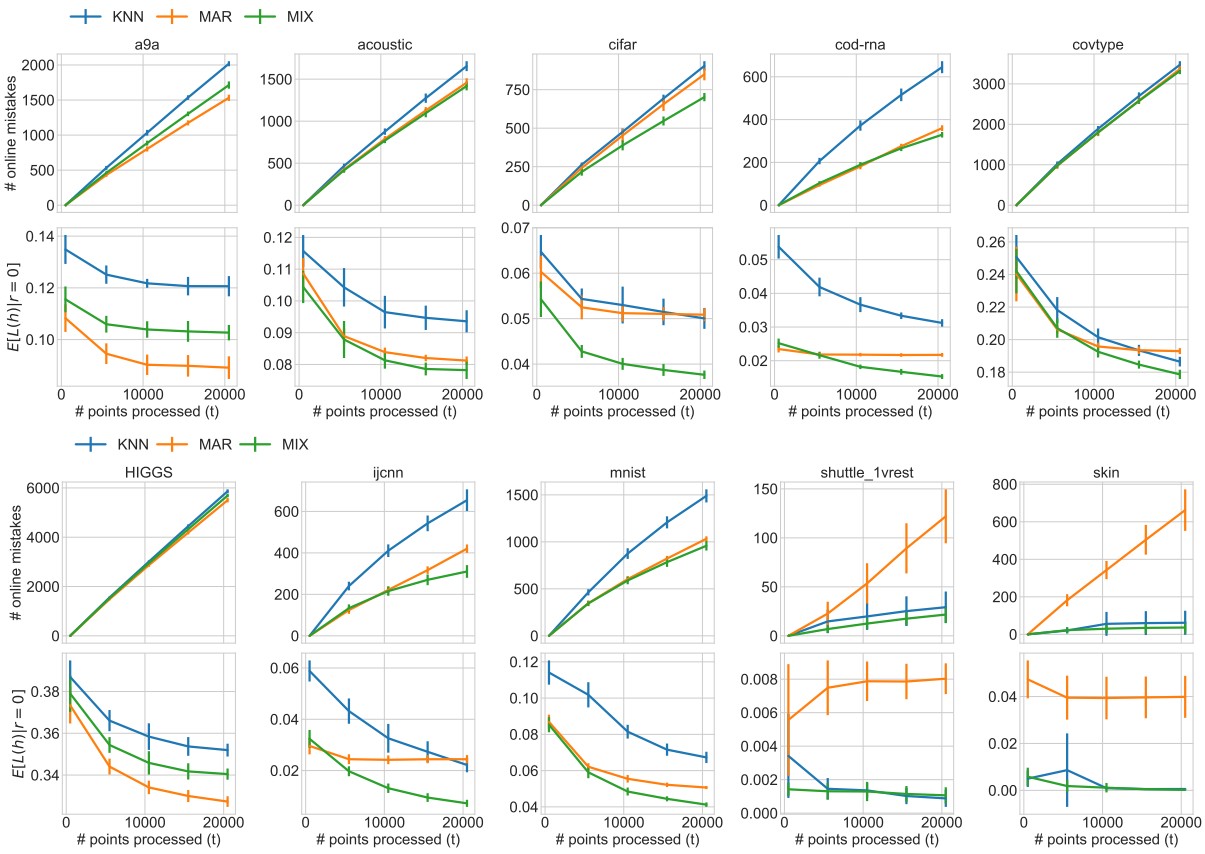

Figure 8: The full learning curves associated to the results presented in Table 2.