# OpenReview forum: "Learning with Labeling Induced Abstentions"
_NeurIPS.cc/2021/Conference — NeurIPS 2021 Poster_

### Official Review · Reviewer_vKGw · 2021-07-16

**Rating:** 6
**Confidence:** 2

**Summary:**

Summary: This paper sets and explores relations between learning and labeling. The article develops a new method requesting labeling combining abstention and active learning. The authors develop a new method and detail their corresponding algorithm DPL-IWAL. They also provide asymptotic statistical guarantees concerning their approach to show the efficacy of their approach (lower and upper bounds).

**Main Review:**

Significance: The results are interesting and motivated. The novelty of the so-called dual purpose learning is perhaps the paper's biggest strength and applications to real-world scenario seem natural and well justified by the ablation study depicted in Section 6 of the article. The paper seems relevant to NeurIPS.

Clarity: I applause the author for the well-written article. Even though, Section 5 and 6 are harder to digest than the rest of the article.
Minor typo: l. 203 "is contains"

Questions/concerns:
1. DP-IWAL requires several hyper parameters and their selection seems obscure. How to set these parameters as the approach is online?
2. I am suprised that the cardinality of the label set $\mathcal{Y}$ does not appear in the upperbounds while it seems natural that the larger the label set, the more labels request are needed. How come the approach remains unchanged between binary classification and extreme classification?


**Time Spent Reviewing:**

2

---

> ### Author Response · Authors · 2021-08-09
> **Comment to Reviewer**
>
> Thank you for your questions and comments. We address them below.
>
> -- DP-IWAL requires several hyper parameters and their selection seems obscure. How to set these parameters as the approach is online?
>
> We note that T (the duration we expect to run the algorithm) and $\rho$ (the budget) are exogenous parameters defined by the problem setting. The $\delta$ (confidence) parameter is usually fixed to small value (e.g. 0.0001) and does not greatly affect the algorithm performance.
>
> If there are other parameters you are referring to, we are happy to clarify further.
>
> -- I am suprised that the cardinality of the label set  Y does not appear in the upperbounds while it seems natural that the larger the label set, the more labels request are needed. How come the approach remains unchanged between binary classification and extreme classification?
>
> Although it is not explicit, the disagreement coefficient does depend on the number of classes and is non-decreasing as the cardinality of the label set increases (because of the supremum).

---

### Official Review · Reviewer_18hs · 2021-07-16

**Rating:** 5
**Confidence:** 3

**Summary:**

The authors propose a new learning problem that combines aspects of active learning and learning with abstention. They formalize the setting, provide an algorithm. They also prove an upper bound on the algorithm’s label complexity and a matching lower bound for any algorithm in this setting. Experiments are presented, too.

**Limitations And Societal Impact:**

adequately addressed

**Main Review:**

In general, I like the setting introduced in the paper, which is practically motivated and nicely combines abstention with active learning. I would have appreciated a better discussion of the consequences of this combination, however, especially on an intuitive level. At first, it looks as if the learner is supposed to find a compromise, but upon closer look, that does not necessarily seem to be the case. From an active learning point of view, it makes sense to query those instances on which the uncertainty is high (except in the case pf pure noise, as used by the authors in their proofs, because nothing can be learned from it), but for the same instances it is reasonable to abstain. So where is the problem, and if the learner cannot achieve the learning rate of the standard (supervised, active) agent, what is its disadvantage? Is it because of the slightly different evaluation?

Upon closer inspection, I was also puzzled by some technical details of the setting. For example, why is a complete request function R_t needed for every t if this function is only applied to a single instance x_t? What the learner needs is a rule that decides whether or not x_t is queried, given the history so far. Why should one consider the conditional accuracy given r(x)=0 and not the overall accuracy, and isn't is a problem that the request function may change from one time point to the other? If eta is the request rate and e the conditional probability of a correct prediction, then the overall accuracy is eta*1 + (1-eta)*e, so maximizing the overall accuracy is equivalent to maximizing e.  Moreover, I was puzzled by the assumption of a finite time horizon T. Is this an online setting or not? Are new examples still queried after T, using r_T, or does querying stop at that time? If it continues, there would not be a need for the finite time horizon. If it stops, that would be a bit strange from a practical point of view.

The paper is quite dense and not always easy to follow, especially since many things are moved to the appendix -- a bit too much in my opinion. For example, I'd have appreciated to see at least a sketch of the proofs in the main paper. They seem to be technically sound, but it's of course hard to check 10 pages of proofs in detail.

The title of the paper is quite unspecific and in my opinion not well chosen.


**Time Spent Reviewing:**

5

---

> ### Author Response · Authors · 2021-08-09
> **Comment to Reviewer**
>
> Thank you for the detailed comments. We hope to resolve some of the questions you had about the technical details of the paper, and look forward to a discussion.
>
> -- In general, I like the setting introduced in the paper ... Is it because of the slightly different evaluation?
>
> Recall that in our setting we have a very specific connection between label requests and model evaluation. In particular, the model is only evaluated on the region where labels are not queried. This is one of the novelties of the setting. In order to see the tension between active learning and abstention, we can examine at least two illustrative examples.
>
> One is in the case of pure noise: as the reviewer is pointing out, an active learner will not want to query points in a pure noise region. However, not querying corresponds to evaluating the model. Hence, the model would be evaluated in the region of pure noise, which is in contrast to a successful abstention strategy, which would abstain in this region. This example is not unique to pure noise, and can be softened scenarios similar in spirit with bounded levels of noise.
>
> A second type of example is illustrated in Appendix C. In this example, there are (non-noisy) regions which our hypothesis class may simply not be able to model and hence a perfect abstainer would select. However, labels from these regions alone would misguide our model training. As a result, and in contrast, a successful active learner algorithm would not query these points.
>
> -- Upon closer inspection, I was also puzzled by some technical details of the setting. For example, why is a complete request function R_t needed for every t if this function is only applied to a single instance x_t? What the learner needs is a rule that decides whether or not x_t is queried, given the history so far.
>
> We believe that these are not materially different approaches. Under the reviewer’s proposal, the decision the learner would have made, for each x that could have been presented at time t, is still a mathematically well-defined quantity. In fact, the algorithm presented in the paper does not actually explicitly define an R_t, and only makes a decision for the x_t presented to it. However, because we want to describe the loss on the region where the algorithm is evaluated (in expectation), for the purpose of analysis, we need the function R_t.
>
> --- Why should one consider the conditional accuracy given r(x)=0 and not the overall accuracy, and isn't is a problem that the request function may change from one time point to the other?  If eta is the request rate and e the conditional probability of a correct prediction, then the overall accuracy is eta*1 + (1-eta)*e, so maximizing the overall accuracy is equivalent to maximizing e.
>
> The formulation suggested by the reviewer makes sense if we are willing to consider systems with perfect accuracy when r(x) = 1, and where the request-rate is constant at eta. Recall our motivation. A system designer wants to understand the quality of some hypothesis h_t only on the region where h_t will be used (when r(x) = 0). This occurs often in real systems. The designer is not capable of modelling accuracy when r(x) = 1, which corresponds to using human-intensive pipelines or expensive testing. Accuracy in this domain might not even be well-defined. They simply want to optimize the ML model’s performance in the domain where it will be executed. A more minor point is that in our setting the budget may not be saturated. At the same time, we do agree that a more holistic approach could be worth investigating, and could make for interesting follow-up work. We discuss this in L23-36, but we will further expand to include this discussion as well.
>
> Regarding the changing request function: our algorithm and our analysis of it, explicitly considers and is valid for an r_t that changes from round to round. We view that formulating as we have done is a contribution of this work.
>
> -- Moreover, I was puzzled by the assumption of a finite time horizon T. Is this an online setting or not? Are new examples still queried after T, using r_T, or does querying stop at that time. If it continues, there would not be a need for the finite time horizon. If it stops, that would be a bit strange from a practical point of view.
>
> Online settings frequently assume finite time-horizons, which we view as an assumption orthogonal to the online versus batch learning scenario.
>
> Our algorithm operates in an online fashion; note that examples arrive one at a time and an immediate decision on whether to query the label or make a prediction must be made. This sequence determines the data available to the algorithm. At the same time, we give an any-time bound on E[L_t(h_t, x_t) | r_t(x) = 0], which characterizes the performance of the model (h_t, r_t) if it were used forever afterwards. We don’t view this as strange from a practical point of view, as it reflects the performance of a model that has been trained up to a certain checkpoint and is then deployed.
>
> -- The paper is quite dense and not always easy to follow, especially since many things are moved to the appendix -- a bit too much in my opinion. For example, I'd have appreciated to see at least a sketch of the proofs in the main paper. They seem to be technically sound, but it's of course hard to check 10 pages of proofs in detail.
>
> Unfortunately we were limited by space, but nonetheless, we will include at least proof sketches in the final version.
>
> -- The title of the paper is quite unspecific and in my opinion not well chosen.
>
> Thank you for your feedback. We went back and forth considering titles that succinctly describe the work. We are happy to take suggestions, and will in any case consider changing the title.

---

### Official Review · Reviewer_iNDa · 2021-07-18

**Rating:** 6
**Confidence:** 2

**Summary:**

The paper introduces a formalization of an online learning setup for labeling with human annotators and machine evaluation and gives an algorithm that simultaneously learns a model and decides when to request a label. Authors prove an upper bound on the algorithm’s label complexity and a matching lower bound for any algorithm adapted to this setup.

**Main Review:**

General Assessment:  The paper is well written and easy to follow. The distinction between proper and improper settings is interesting and the lower bound result for proper setting is novel and relevant.

Comments:
- Empirical comparison to existing approaches in active learning is missing. Are there methods in active learning that can be adapted and compared to the proposed DPL-IWAL algorithm.
- Is it possible to make proper dual-purpose labeling framework work in an online setting with some distributional/margin assumptions. It would be interesting to see how the lower bound can be avoided in a typical practical scenario?


**Time Spent Reviewing:**

4

---

> ### Author Response · Authors · 2021-08-09
> **Comment to Reviewer**
>
> Thank you for your questions and comments. We address them below.
>
> -- Empirical comparison to existing approaches in active learning is missing. Are there methods in active learning that can be adapted and compared to the proposed DPL-IWAL algorithm.
>
> In our experiments, we compare our method to an active learning algorithm called margin, also known as uncertainty sampling. Please see lines 371-373. Out of the possible active learning algorithms, we decided to compare against margin sampling since it is an algorithm that admits a substantial improvement in practice compared to other active learning algorithms and it is favored by practitioners. Moreover, the margin algorithm is often used as a competitive comparator in recent active learning papers (see for example Ash et al. “Deep Batch Active Learning by Diverse, Uncertain Gradient Lower Bounds” ICLR. 2020).
>
> --Is it possible to make proper dual-purpose labeling framework work in an online setting with some distributional/margin assumptions. It would be interesting to see how the lower bound can be avoided in a typical practical scenario?
>
> The focus of this paper is on deriving bounds for the most general agnostic setting, but one can indeed add additional distributional assumptions. A Massart noise assumption (or something similar) would exclude the lower bound described in the paper, and is an interesting direction for future work.

---

### Decision · Program_Chairs · 2021-09-27

**Decision:**

Accept (Poster)

**Comment:**

This work formally proposes a novel learning  setup, which, on a high level can be viewed as a combination of active learning and learning with abstentions. The framework works with two function classes, a function class R, that contains regions of where labels should be requested and a (standard) hypothesis class of classifiers H. The goal of the earning is set to minimize the classification loss in the regions where labels are not requested, that is, minimize classification loss conditioned on r=0 (in the regions where r=1, labels are assumed to be always requested, so the classifier does not need to get labels correct). To make this setup non trivial, the the region, where r=1 is budgeted to probability weight rho.

This paper proposes and analyzes this setup. The submission contains upper and lower bound on the query complexity and this is compared with standard active learning. Additionally, the authors provide some experimental evaluation on an adaptation of IWAL to their setup.

This overall appears to be a solid and rounded piece of work. The reviewers have critized a lack of proper motivation as well as clarity issues in the presentation. More intuitive explanation and guidance to the readers could have been given. I agree with the reviewers on these aspects, and would therefore rate this work as borderline, with a storng tendency to accept.